# Hormetic heat stress and HSF-1 induce autophagy to improve survival and proteostasis in *C. elegans*

Caroline Kumsta[1], Jessica T. Chang[1], Jessica Schmalz[1] & Malene Hansen[1]

Stress-response pathways have evolved to maintain cellular homeostasis and to ensure the survival of organisms under changing environmental conditions. Whereas severe stress is detrimental, mild stress can be beneficial for health and survival, known as hormesis. Although the universally conserved heat-shock response regulated by transcription factor HSF-1 has been implicated as an effector mechanism, the role and possible interplay with other cellular processes, such as autophagy, remains poorly understood. Here we show that autophagy is induced in multiple tissues of *Caenorhabditis elegans* following hormetic heat stress or HSF-1 overexpression. Autophagy-related genes are required for the thermo-resistance and longevity of animals exposed to hormetic heat shock or HSF-1 overexpression. Hormetic heat shock also reduces the progressive accumulation of PolyQ aggregates in an autophagy-dependent manner. These findings demonstrate that autophagy contributes to stress resistance and hormesis, and reveal a requirement for autophagy in HSF-1-regulated functions in the heat-shock response, proteostasis and ageing.

[1] Development, Aging, and Regeneration Program, Sanford Burnham Prebys Medical Discovery Institute, 10901 North Torrey Pines Road, La Jolla, California 92037, USA. Correspondence and requests for materials should be addressed to M.H. (email: mhansen@sbpdiscovery.org).

Organisms have developed highly regulated stress-response pathways to combat exogenous and endogenous stresses, and maintain cellular homeostasis. In response to environmental stresses such as increased temperature, the conserved transcription factor HSF-1 binds to heat shock elements (HSEs)[1] in the promoters of heat-inducible genes and induces expression of heat shock proteins (HSPs) and molecular chaperones. These proteins detect and refold unfolded or misfolded proteins and prevent their accumulation, a process known as the heat shock response (HSR)[2]. HSF-1 is essential for maintaining proteostasis and can suppress protein toxicity and aggregation in several organisms[3–6]. Proteotoxicity and protein misfolding increase with age and contribute to a number of late-onset neurodegenerative diseases[7,8]. For example, Huntington's disease is caused by the presence of an expansion of a polyglutamine (PolyQ) tract in the protein huntingtin, which makes it prone to aggregation. Age-dependent increases in proteotoxicity can be modelled in the nematode Caenorhabditis elegans, in which aggregation of PolyQ-containing proteins and other metastable proteins begins at the onset of adulthood[3,5,9,10]. In C. elegans, this increase in proteotoxicity is accompanied by a decline in proteostasis networks, including the HSR[11–13]. This repression of the HSR can however be manipulated by overexpression of HSF-1, which diminishes the proteotoxicity of several aggregation-prone proteins into adulthood[5,10]. In addition to improving proteostasis, HSF-1 overexpression also increases longevity and improves stress resistance in C. elegans[3,6]. Although reduction of several HSF-1 target genes, such as the molecular chaperones hsp-16.1, hsp-16.49, hsp-12.6 and sip-1, partially reduces the longevity conferred by HSF-1 overexpression[3], other effectors of HSF-1-mediated longevity have yet to be identified.

Similar to HSF-1 overexpression, hormetic stress can also increase lifespan and stress resistance. The concept of hormesis refers to a beneficial low-dose stimulation with an environmental agent or exposure to an external stressor that is toxic at a high dose[14,15]. This phenomenon has been observed in many species, including C. elegans[16–18], the fruit fly Drosophila[19–22] and human fibroblasts[23]. In Drosophila[24] and human fibroblasts[25], hormetic heat shock upregulates HSF-1 target genes such as the molecular chaperone hsp-70 and in C. elegans the expression levels of another HSF-1 target gene hsp-16.2, following heat shock, can be used to predict lifespan[26]. Although it has been suggested that hormesis occurs through the activation of stress-response pathways, including HSR/HSF-1 and insulin/insulin growth factor-1 signalling[14,27], it remains unclear whether HSPs are the only effector molecules of hormesis or whether other proteostatic determinants are similarly important.

Macroautophagy (hereafter referred to as autophagy) is another cytoprotective mechanism that plays a major role in cellular homeostasis. Autophagy facilitates degradation and recycling of cytosolic components in response to stresses such as nutrient deprivation, hypoxia, cytotoxic chemicals and pathogens[28]. Autophagy is initiated by the nucleation of a double membrane, which elongates into an autophagosomal vesicle that encapsulates cytoplasmic material, including damaged macromolecules and organelles. Subsequent fusion of autophagosomes and lysosomes leads to formation of autolysosomes, in which the sequestered contents are degraded by hydrolases and recycled. This sequential process is governed by conserved proteins encoded by autophagy (ATG)-related genes. Of note, lipid-bound Atg8, which inserts into the membrane of autophagosomes and is important for their formation, can be expressed as a green fluorescent protein (GFP)-tagged protein for use as a marker of autophagosome abundance[29]. In C. elegans, many longevity paradigms have been shown to increase autophagy markers and require autophagy genes for their long lifespan[30]. Notably, many of these longevity models also require hsf-1 (ref. 31). Heat shock can modulate autophagy in several cell models and the HSF-1-regulated HSR and autophagy may be coordinated under certain stress conditions (reviewed in ref. 32); however, it remains unclear how autophagy contributes to stress resistance in organisms subjected to stressors such as hormetic heat shock.

Here we sought to elucidate the molecular mechanisms underlying the beneficial effects of hormetic heat stress by investigating the interplay between heat shock, HSF-1 and autophagy in C. elegans. Hormetic heat shock and HSF-1 overexpression induce autophagy in multiple tissues of C. elegans and autophagy-related genes are essential for both heat shock-induced and HSF-1-mediated stress resistance and longevity. Finally, we find that hormetic heat shock also improves several models of protein aggregation in an autophagy-dependent manner. These observations are important, because they indicate that autophagy induction by hormetic heat stress is an important mechanism to enhance proteostasis, possibly also in age-related protein-folding diseases.

## Results

**Hormetic heat shock induces autophagy in C. elegans.** Exposure of C. elegans to hormetic heat shock early in life increases their survival[15,16], but the molecular mechanisms underlying the hormetic benefits are not well understood. To better understand the molecular mechanisms engaged in organisms subjected to hormetic heat shock, we examined C. elegans responses using a hormetic heat shock regimen of 1 h at 36 °C on day 1 of adulthood. This treatment promotes C. elegans survival[16–18] (Supplementary Tables 1 and 2) and selectively induced the HSR, as shown by the marked induction of HSP genes such as hsp-70 and hsp-16.2, and only modestly induced the mitochondrial stress gene hsp-6 and the oxidative stress gene gst-4, whereas other oxidative or endoplasmic reticulum stress-response markers were not induced (Supplementary Fig. 1).

We monitored autophagy in individual tissues of C. elegans subjected to hormetic heat shock by expressing a GFP-tagged LGG-1/Atg8 reporter[33,34], which allows autophagosomes to be visualized as fluorescent punctae. We detected an increase in the number of GFP::LGG-1/Atg8 punctae in all tissues examined in heat-shocked animals; namely, hypodermal seam cells (Fig. 1a), striated body-wall muscle cells (Fig. 1b), neurons located in the nerve ring (Fig. 1c) and proximal intestinal cells (Fig. 1d and see also Supplementary Table 3). These punctae represented autophagosomal structures and not heat shock-induced GFP aggregates, as we did not observe punctae formation in any tissues upon heat shock in C. elegans expressing a mutant form of GFP-tagged LGG-1/Atg8 protein (G116A) that is defective in lipidation and autophagosome targeting (Supplementary Fig. 2 and Supplementary Table 4)[35]. This was further confirmed by the reduction of GFP::LGG-1/Atg8 punctae by RNA interference (RNAi)-mediated silencing of multiple autophagy genes (Supplementary Fig. 3 and Supplementary Table 5). Of note, autophagy gene RNAi did not compromise the organism's ability to induce the HSR, as neither the induction of the reporter genes hsp-16.2 and hsp-70 (Supplementary Figs 4a,b) nor the expression of HSP genes hsp-16.1 and aip-1 (Supplementary Fig. 4c) was affected after autophagy gene reduction. These data demonstrate that animals with reduced levels of autophagy genes still have the capacity to induce a HSR.

The observed increases in autophagosome abundance following heat shock could result from enhanced autophagy induction

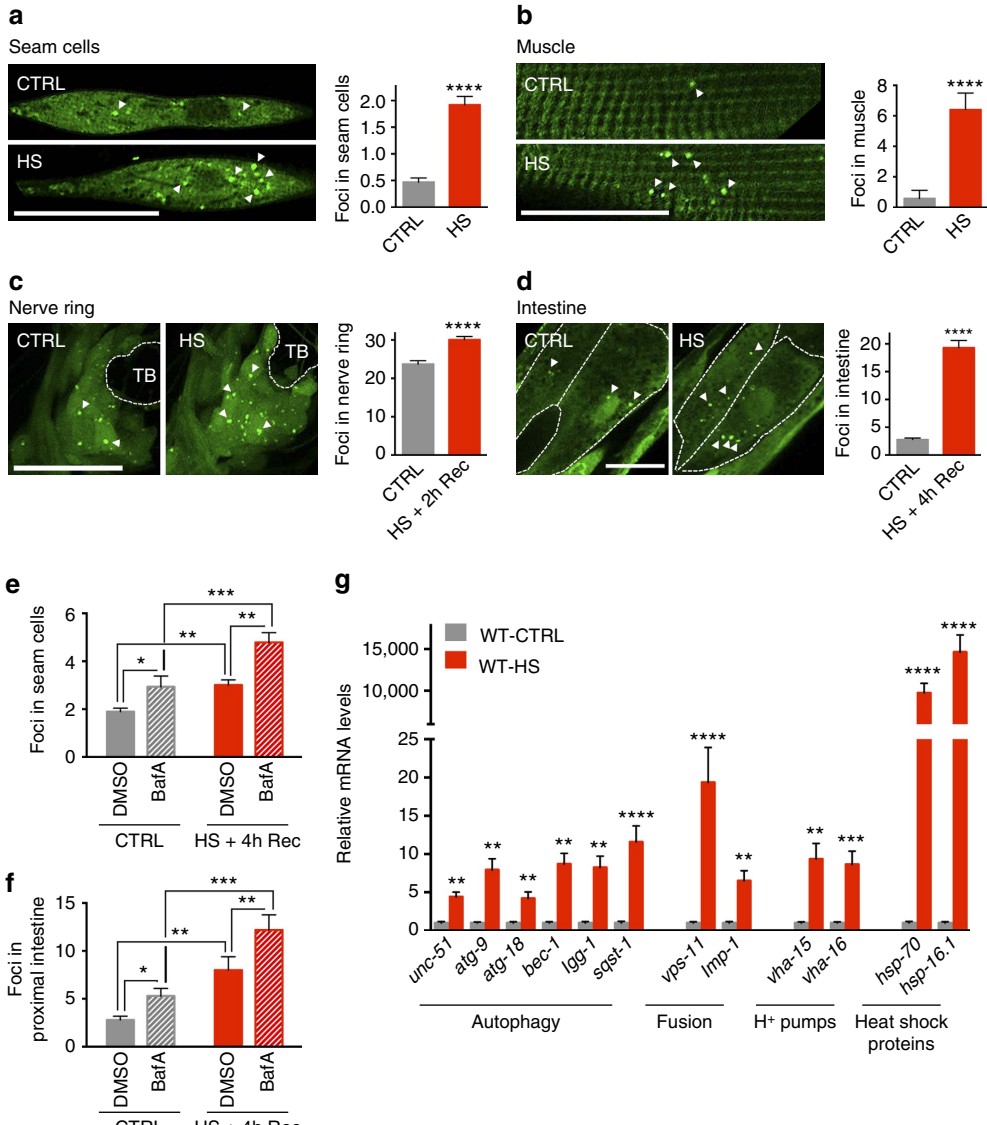

**Figure 1 | Heat shock induces autophagy.** (**a**–**d**) GFP::LGG-1/Atg8 punctae were counted in wild-type *C. elegans* maintained under control conditions (CTRL) or subjected to heat shock for 1 h at 36 °C (HS) followed by the indicated recovery period (Rec). Punctae were examined in (**a**) hypodermal seam cells (N = 63-98 cells), (**b**) body wall muscle (N = 10–12 animals), (**c**) nerve ring neurons (N = 12 animals) and (**d**) proximal intestinal cells (N = 14–16 animals). See also Supplementary Table 3 for a summary of repeat experiments. (**e,f**) Autophagy-flux measurements were performed on day 1 of adulthood in animals maintained at 15 °C (CTRL) or subjected to heat shock for 1 h at 36 °C (HS) followed by injection with vehicle (DMSO) or bafilomycin A (BafA) to block autophagy at the lysosomal acidification step. The number of GFP::LGG-1/Atg8 punctae was counted in (**e**) hypodermal seam cells (N = 28-39, n = 2) and (**f**) the proximal intestine (N = 14-17, n = 2). (**g**) Transcript levels of genes involved in various steps of the autophagy process in wild-type (WT) animals maintained under control conditions (CTRL) or subjected to heat shock for 1 h at 36 °C (HS). Data are the mean ± s.e.m. of four biological replicates, each with three technical replicates, and are normalized to the mean expression levels of four housekeeping genes. All error bars are s.e.m. Scale bars, 20 μm. TB, terminal pharyngeal bulb. *$P < 0.05$, **$P < 0.01$, ***$P < 0.001$, ****$P < 0.0001$ by Student's *t*-test (**a**–**d**), two-way ANOVA (**e,f**) and multiple *t*-tests (**g**).

or from inhibition of autophagosome turnover. To distinguish between these possibilities, we used bafilomycin A (BafA), a chemical inhibitor of autophagy, to assess autophagy flux in the hypodermal seam cells and the intestine[36,37]. BafA blocks autophagosomal turnover by inhibiting V-ATPase activity and preventing lysosomal acidification[29]. A change in the number of autophagosomes upon BafA addition thus indicates that autophagy is active, whereas no change indicates that the cell/tissue is experiencing a block in autophagy. We found that BafA treatment increased the number of GFP::LGG-1/Atg8 punctae in the hypodermal seam cells and the intestines of

heat-shocked animals (Fig. 1e–f), indicating that the increase in autophagosomes in these tissues represents an induction of autophagy rather than inhibition of autophagosome turnover.

To further characterize the tissue-specific autophagy induction upon heat shock, we performed time-course experiments in hypodermal seam cells, body-wall muscle, nerve-ring neurons and intestinal cells. Specifically, we exposed animals to heat shock for 1 h and then monitored the abundance of GFP::LGG-1/Atg8 punctae for a total of 30 h of recovery time. In the hypodermal seam cells and body-wall muscles, the number of autophagic punctae began to increase immediately after heat

shock, whereas the response was delayed 2–4 h in both the nerve ring and the intestine (Supplementary Fig. 5). Furthermore, the autophagy response was transient in the body-wall muscle (~4 h duration) and the nerve ring (~6 h), but was sustained in the intestine (~12 h) and hypodermal seam cells (~30 h) (Supplementary Fig. 5 and Supplementary Table 3). We also observed tissue-specific differences in the duration of heat shock required to induce autophagy. Exposure to elevated temperatures for 15 min was sufficient to cause a near-maximal increase in GFP::LGG-1/Atg8 punctae in the hypodermal seam cells, body-wall muscle and nerve ring, whereas the intestine required at least 45 min of heat shock to display a significant increase in punctae (Supplementary Fig. 6 and Supplementary Table 6). Collectively, these results demonstrate that individual tissues display distinct autophagic responses to heat stress.

**Heat shock leads to increased autophagy gene expression**. Although autophagy is subject to extensive posttranslational regulation[38], recent studies have indicated an important role for transcriptional regulation of autophagy genes in homeostatic adaptation[30]. Therefore, we analysed the expression levels of autophagy-related genes in wild-type C. elegans exposed to heat shock. As expected, this treatment markedly increased (>1000-fold) expression of the HSP genes hsp-70 (C12C8.1) and hsp-16.1. In addition, we observed a 5–20-fold induction of multiple autophagy genes, including those involved in autophagosome formation, autophagosome–lysosome fusion and lysosomal degradation (Fig. 1g). Heat shock also induced transcriptional reporters of several autophagy genes, including the phosphoinositide-binding protein atg-18, the SQSTM1/p62 orthologue sqst-1 and atg-16.2, which is involved in phagophore formation (Supplementary Fig. 7). Taken together, these data indicate that autophagy-related genes are transcriptionally upregulated in response to a hormetic heat shock.

**Overexpression of HSF-1 is sufficient to induce autophagy**. As we observed increased expression of autophagy genes upon heat shock and because the benefits of hormesis are at least partially mediated by the activation of the HSR regulated by HSF-1 (ref. 39), we further investigated a role for the transcription factor HSF-1 in autophagy regulation. We monitored autophagy in the tissues of animals overexpressing HSF-1 (ref. 40) and detected an increase in GFP::LGG-1/Atg8 punctae in hypodermal seam cells (Fig. 2a), body-wall muscles (Fig. 2b), nerve ring neurons (Fig. 2c) and proximal intestinal cells (Fig. 2d and see also Supplementary Table 7). Moreover, injections of BafA increased the number of GFP::LGG-1/Atg8 punctae in hypodermal seam cells (Fig. 2e) and the intestine (Fig. 2f) of animals overexpressing HSF-1, and expression of autophagy-related genes was much higher in animals overexpressing HSF-1 than in wild-type animals under basal (non-stressed) conditions (Fig. 2g). Taken together, these observations suggest that HSF-1 overexpression alone is sufficient to induce autophagy, thus recapitulating the effects of heat shock on wild-type animals (Fig. 1). RNAi-mediated silencing of hsf-1 in wild-type animals prevented the heat shock-induced increase in GFP::LGG-1/Atg8 punctae in body-wall muscles, nerve ring neurons and proximal intestinal cells (hypodermal seam cells were an exception, see Supplementary Fig. 8 and Supplementary Table 5), consistent with hsf-1 being required for a heat shock-dependent increase of autophagosomes at least in these three major tissues. Collectively, our results suggest that HSF-1 regulates autophagy in C. elegans. Although many of the autophagy-related genes that were induced upon heat shock contain at least one putative HSE in their promoter regions (Supplementary Table 8), it remains to be

determined whether HSF-1 regulates autophagy directly or whether other transcriptional regulators besides HSF-1 may be involved in the upregulation of autophagy genes upon heat shock.

In support of the latter possibility, we found that hlh-30, the orthologue of mammalian transcription factor EB and a conserved regulator of multiple autophagy-related and lysosomal genes[30], was required for the induction of several autophagy genes upon heat shock (Supplementary Fig. 9a). Moreover, hormetic heat shock caused a rapid translocation of GFP-tagged HLH-30 to the nucleus in multiple tissues, including the nerve ring, pharynx, vulva, tail and intestine (Supplementary Fig. 9b,c), indicating a possible activation of HLH-30 (refs 41–43). These observations therefore point to a role for HLH-30 in regulating heat shock-induced autophagy. It will be interesting to investigate how HSF-1 and HLH-30 coordinate their effects on autophagy gene expression and the extent to which direct or indirect regulatory mechanisms are involved.

**Autophagy genes are required for heat shock-mediated survival**. Our results demonstrate that exposure of C. elegans to hormetic heat shock early in life (day 1 of adulthood) induces autophagy. Therefore, we next asked whether autophagy gene expression is required to observe the long-term health benefits of hormetic heat stress. Animals subjected to mild heat stress on day 1 of adulthood were indeed more resistant to thermal stress later in life (day 4 and 5 of adulthood) (Fig. 3a and Supplementary Table 1)[16]. Importantly, this resistance was significantly reduced by RNAi-mediated suppression of genes involved in multiple aspects of autophagy, including autophagy initiation (unc-51/ATG1), membrane nucleation (bec-1/ATG6), phosphoinositide 3-phosphate binding (atg-18) and autophagosome elongation (lgg-1/ATG8) (Fig. 3a and Supplementary Tables 1), suggesting that functional autophagy is required for the beneficial effect of hormetic heat stress on thermotolerance.

Hormetic heat stress also increases longevity in C. elegans (Fig. 3b–d and reviewed in ref. 44). Here, too, we found that autophagy genes (unc-51/ATG1, bec-1/ATG6, lgg-1/ATG8, atg-18 and atg-13; the latter two involved in phagophore formation) were required for the increased lifespan of wild-type animals exposed to hormetic heat shock early in life (Fig. 3b–d and Supplementary Table 2). Similarly, hlh-30 was required for the beneficial effects of hormetic heat shock on thermal stress resistance (Supplementary Fig. 9d and Supplementary Table 9) and longevity (Supplementary Fig. 9e and Supplementary Table 10). These data therefore strongly support a role for autophagy genes and hlh-30 in mediating the beneficial effects of hormetic heat shock on stress resistance and longevity in C. elegans.

Previous work has shown that heat shock for 1–4 h can dramatically reduce pharyngeal pumping in C. elegans[45]. We also observed a rapid decline in pharyngeal pumping during the hormetic heat shock, which was fully reversed within 30 min of returning the animals to 20 °C (Supplementary Fig. 10a). We ruled out that the stress resistance and longevity observed after a hormetic heat shock was due to dietary restriction caused by decreased pharyngeal pumping, as animals that were dietary restricted for 90 min at day 1 of adulthood (either in liquid media or on agarose plates) showed no hormetic benefits (Supplementary Figs 10b,c and Supplementary Tables 1 and 2).

Finally, to determine whether the observed benefits of hormetic heat shock could also be experienced by animals later in life, we heat shocked the animals on days 1, 3, 5 or 7 of adulthood and analysed their thermorecovery 2 days later. Exposure on days 1, 3 and 5 of adulthood increased the subsequent stress

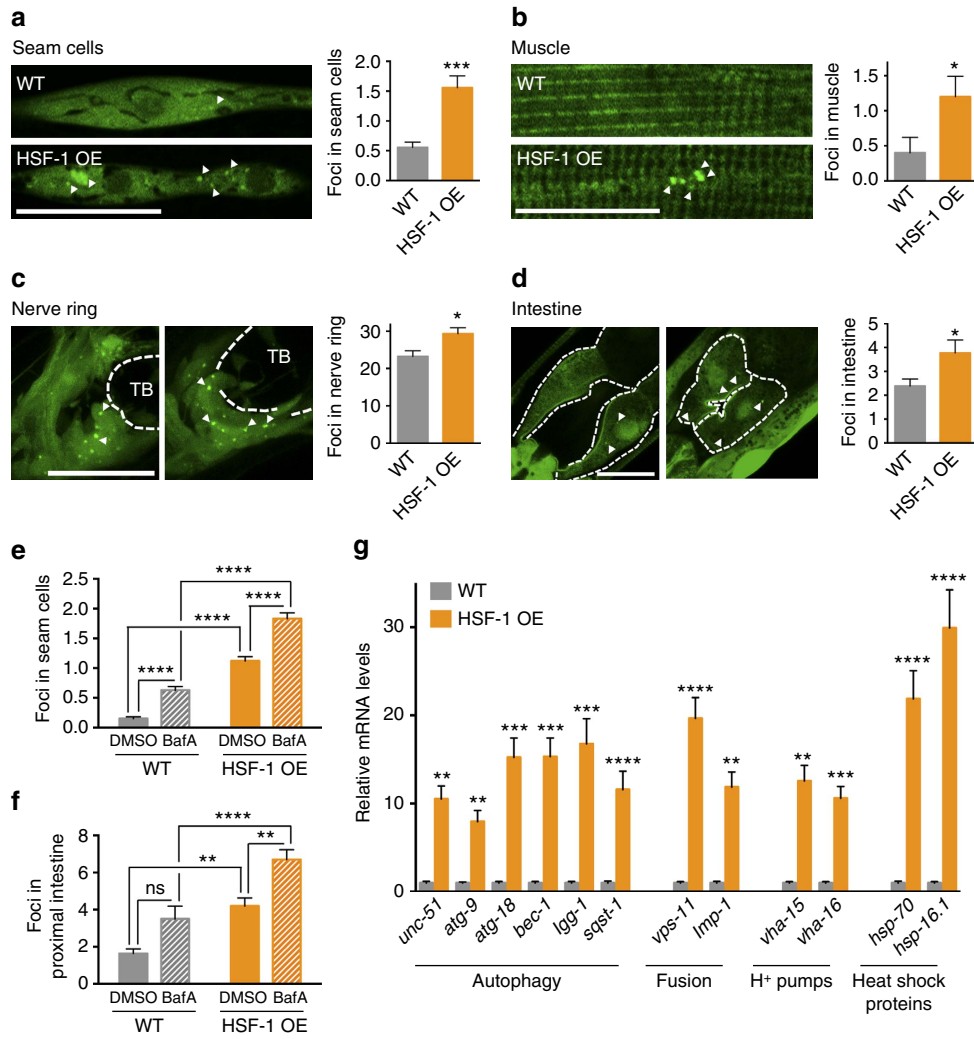

**Figure 2 | Autophagy is induced in HSF-1-overexpressing animals.** (**a**–**d**) GFP::LGG-1/Atg8 punctae were counted in (**a**) hypodermal seam cells ($N = 131$–163 cells), (**b**) body-wall muscle ($N = 10$ animals), (**c**) nerve-ring neurons ($N = 11$–12 animals) and (**d**) proximal intestinal cells ($N = 13$ animals) of wild-type (WT) and HSF-1-overexpressing (HSF-1 OE) animals. See also Supplementary Table 7 for a summary of repeat experiments. (**e**,**f**) Autophagy-flux measurements were performed on day 1 of adulthood in animals maintained at 20 °C. WT and HSF-1 OE animals were injected with vehicle (DMSO) or bafilomycin A (BafA) to block autophagy at the lysosomal acidification step. The number of GFP::LGG-1/Atg8 punctae was counted in (**e**) hypodermal seam cells ($N = 129$–162, $n = 3$) and (**f**) the proximal intestine ($N = 21$–26, $n = 3$). (**g**) Transcript levels of genes involved in various steps of the autophagy process in WT and HSF-1 OE animals. Data are the mean ± s.e.m. of four biological replicates, each with three technical replicates, and are normalized to the mean expression levels of four housekeeping genes. All error bars are s.e.m. Scale bars, 20 µm. TB, terminal pharyngeal bulb. ns: $P > 0.05$, *$P < 0.05$, **$P < 0.01$, ***$P < 0.001$ and ****$P < 0.0001$ by Student's $t$-test (**a**–**d**), two-way ANOVA (**e**,**f**) and multiple $t$-tests (**g**).

resistance and lifespan, with the greatest effect on day 1; however, there was no significant effect on day 7 (Supplementary Fig. 11a,b and Supplementary Tables 1 and 2). As previously reported, the ability of animals to respond to a hormetic treatment decreased with age[46] (Supplementary Fig. 11a,b and Supplementary Tables 1 and 2). In addition, we measured the messenger RNA levels of HSP and autophagy genes in animals that were heat shocked later in life. HSP genes *hsp-70* and *hsp-16.1*, and autophagy genes *bec-1*/ATG6 and *sqst-1*/SQSTM1/p62 were heat inducible on days 1 through 7 of adulthood (with a slight age-associated reduction in magnitude), and *atg-18* and *lgg-1*/ATG8 were inducible only in animals heat shocked on day 1 of adulthood. These findings are consistent with the notion that both autophagic activity and the beneficial effects of hormetic heat stress decline with age[47,48].

**Autophagy genes are required for HSF-1-mediated survival.** As HSF-1 overexpression in *C. elegans* is sufficient to induce autophagy (Fig. 2) and increase stress resistance and longevity[3] (Supplementary Tables 11 and 12), we asked whether autophagy genes were also required for the increased thermotolerance and longevity[3,6] observed in animals overexpressing HSF-1. Indeed, silencing of autophagy genes reduced the enhanced stress tolerance (Fig. 3e and Supplementary Table 11) and longevity (Fig. 3f–h and Supplementary Table 12) conferred by HSF-1 overexpression. These findings indicate that autophagy is essential for resistance to thermal stress and lifespan extension of animals overexpressing HSF-1.

**Heat shock and HSF-1 improve proteostasis via autophagy.** An important hallmark of organismal ageing is the loss of proteostasis. A key example is the age-associated increase in aggregation of disease-related factors such as PolyQ-containing proteins[9], which can cause neurodegenerative disorders such as Huntington's disease[7]. PolyQ proteins and metastable proteins

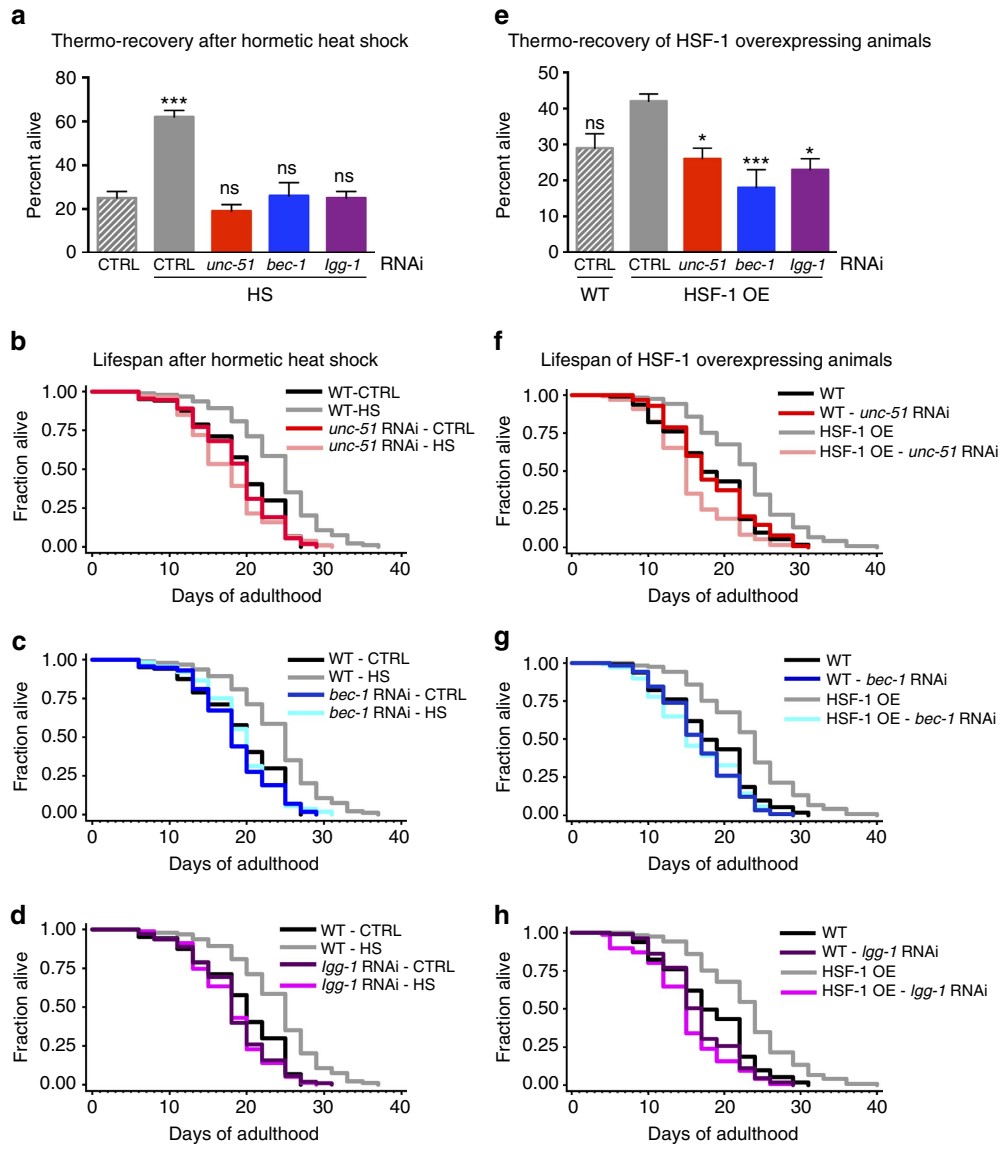

**Figure 3 | Autophagy genes are required for heat shock- and HSF-1-mediated survival.** (**a**) Survival of wild-type (WT) animals subjected to hormetic heat shock on day 1 of adulthood and then incubated for 8 h at 36 °C on day 4 of adulthood. Animals were fed from day 1 of adulthood with control bacteria (empty vector, CTRL) or bacteria expressing dsRNA targeting the autophagy genes *unc-51/ATG1*, *bec-1/ATG6* and *lgg-1/ATG8* (N = 65–90 animals, n = 4 plates). (**b–d**) Lifespan analysis of animals subjected to hormetic heat shock with RNAi-mediated autophagy gene reduction from day 1 of adulthood. WT-CTRL animals (19.2 days, N = 104) compared with WT-HS animals (23.7 days, N = 94): P < 0.0001, (**b**) *unc-51/ATG1* RNAi-CTRL (18.5 days, N = 110) compared with *unc-51/ATG1* RNAi-HS (17.5 days, N = 107): P = 0.04, (**c**) *bec-1/ATG6* RNAi-CTRL (19.2 days, N = 116) compared with *bec-1/ATG6* RNAi-HS (18.4 days, N = 112): P = 0.3, (**d**) *lgg-1/ATG8* RNAi-CTRL (18.1 days, N = 108) compared with *lgg-1/ATG8* RNAi-HS (17.9 days, N = 79): P = 0.7. (**e**) Survival of WT or HSF-1-overexpressing (HSF-1 OE) animals incubated for 8 h at 36 °C on day 3 of adulthood. Animals were fed from day 1 of adulthood with control bacteria (empty vector, CTRL) or bacteria expressing dsRNA targeting the indicated autophagy genes (N = 113–220 animals, n = 4 plates). Error bars indicate s.e.m. ns: P > 0.05, *P < 0.05 and ***P < 0.001 by one-way ANOVA. (**f–h**) Lifespan analysis of WT and HSF-1 OE animals subjected to RNAi-mediated autophagy gene reduction from day 1 of adulthood. WT animals (18.1 days, N = 113) compared with HSF-1 OE animals (23.0 days, N = 121): P < 0.0001, (**f**) WT: CTRL compared with *unc-51/ATG1* RNAi (18.3 days, N = 128): P = 0.9, HSF-1 OE: CTRL compared with *unc-51/ATG1* RNAi (15.4 days, N = 133): P < 0.0001, (**g**) WT: CTRL compared with *bec-1/ATG6* RNAi (16.7 days, N = 123): P = 0.02, HSF-1 OE: CTRL compared with *bec-1/ATG6* RNAi (16.3 days, N = 140): P < 0.0001, (**h**) WT: CTRL compared with *lgg-1/ATG8* RNAi (16.7 days, N = 109): P = 0.02, HSF-1 OE: CTRL compared with *lgg-1/ATG8* RNAi (14.9 days, N = 147): P < 0.0001, by log-rank test. See Supplementary Tables 1, 2, 11 and 12 for details on thermorecovery and lifespan analyses and replicate experiments.

expressed in *C. elegans* can model protein-folding diseases and serve as protein-folding sensors[49]. Given the beneficial hormetic effects of heat shock on thermoresistance and longevity, we asked whether hormetic heat shock could also improve proteostasis, as has been observed for HSF-1 overexpression in a muscle PolyQ model[3]. For this, we examined *C. elegans* expressing Q44::YFP

specifically in the intestine[50], the tissue in which GFP::LGG-1/Atg8 punctae were most increased by heat shock (Fig. 1d). Exposure of these animals to hormetic heat shock on day 1 of adulthood significantly reduced the number of intestinal PolyQ44 aggregates on days 2–5 (Fig. 4a–c and Supplementary Fig. 12). Heat shock on day 1 also reduced aggregate accumulation in a

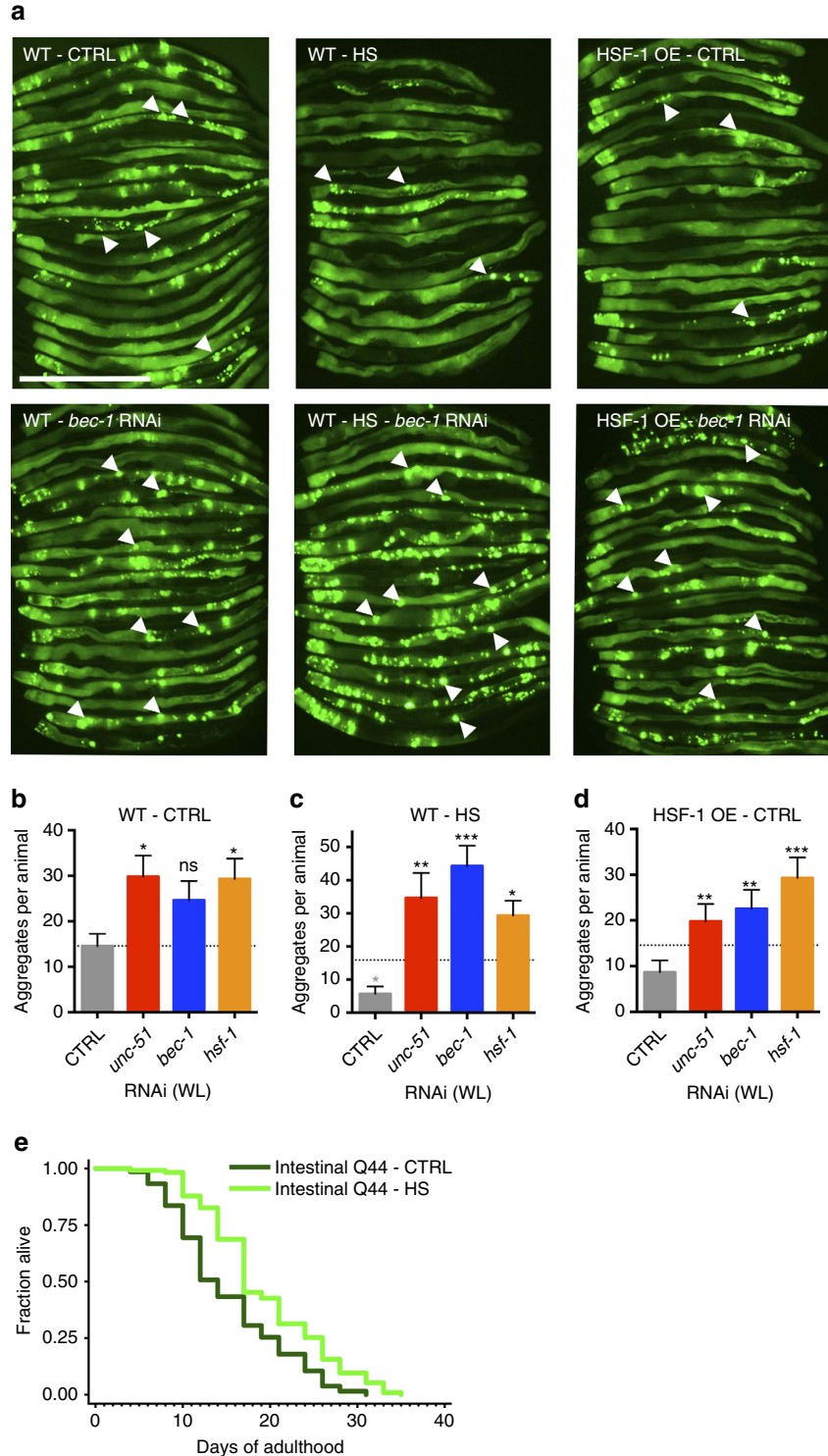

**Figure 4 | Hormetic heat shock reduces PolyQ protein aggregation.** (**a**) Intestinal PolyQ aggregates detected on day 5 of adulthood in wild-type (WT) animals maintained under control conditions (WT-CTRL) or subjected to hormetic heat shock (1h at 36 °C) on day 1 of adulthood (WT-HS) and in HSF-1-overexpressing animals maintained under control conditions (HSF-1 OE-CTRL). Animals expressing PolyQ44::YFP under the control of the intestine-specific promoter *vha-6* were fed from hatching with control bacteria (empty vector; upper row) or bacteria expressing dsRNA targeting *bec-1/ATG6* RNAi (lower row). Arrowheads indicate prominent aggregates. Scale bar, 500 μm. (**b–d**) Quantification of PolyQ aggregates on day 5 of adulthood in animals fed from hatching (whole life, WL) with control bacteria (empty vector, CTRL) or bacteria expressing dsRNA targeting *unc-51/ATG1*, *bec-1/ATG6* or *hsf-1* in (**b**) WT animals, (**c**) WT animals subjected to hormetic heat shock on day 1 of adulthood and (**d**) HSF-1 OE animals (N = 14–23). Dotted line represents number of aggregates of WT-CTRL on empty vector control and grey asterisk represents P-value compared to WT-CTRL on empty vector. The experiments were repeated at least three times with similar results. (**e**) Lifespan analysis of animals expressing intestinal PolyQ44::YFP and subjected to hormetic heat shock on day 1 of adulthood. Intestinal Q44-CTRL animals (15.2 days) compared with Intestinal Q44-HS animals (19.3 days): P < 0.0001, see Supplementary Table 13 for details on lifespan analyses and replicate experiments. Error bars indicate s.e.m. ns: P > 0.05, *P < 0.05, **P < 0.01 and ***P < 0.001 by one-way ANOVA (**b–d**) and log-rank test (**e**).

neuronal PolyQ strain[51] on day 7 of adulthood (Supplementary Fig. 13a,b), suggesting that the hormetic heat stress paradigm protects against PolyQ aggregation in multiple tissues. Notably, the reduction in PolyQ aggregates in intestinal cells and neurons significantly increased the animals' lifespan (~20%; Fig. 4e, Supplementary Fig. 13c and Supplementary Table 13). Thus, the hormetic heat shock improved both proteostasis and lifespan. PolyQ aggregation in the intestine, which began as early as day 1 of adulthood, modestly induced transcription of HSP and autophagy-related genes, although this reached significance only for the HSP genes on day 1 of adulthood (Supplementary Fig. 14a,b). In contrast, neither HSP nor autophagy genes were induced in animals expressing neuron-specific PolyQ proteins (Supplementary Fig. 14a,b), perhaps because protein aggregation was limited in these animals at the time points examined. Importantly, hormetic heat shock significantly induced autophagy-related gene expression in both strains of PolyQ-expressing animals (Supplementary Fig. 14c,d), suggesting that induction of autophagy may contribute to the beneficial effects of hormetic heat shock on proteostasis in these animals.

To directly test this, we subjected PolyQ-expressing animals to autophagy gene RNAi and monitored the effect of hormetic heat stress on aggregation formation. Notably, unc-51/ATG1 and bec-1/ATG6 RNAi abolished the beneficial effect of hormetic heat stress on intestinal and neuronal PolyQ aggregation (Fig. 4a–c and Supplementary Figs 12 and 13), indicating that autophagy is required for the proteostatic benefits of hormetic heat shock. We also found that autophagy genes were required for proteostasis in other tissue-specific aggregation models caused by protein misfolding, as inhibition of autophagy genes enhanced paralysis and movement defects in animals harbouring temperature-sensitive missense mutations in genes affecting the function of paramyosin and myosin (muscle) (Supplementary Fig. 15a,b), dynamin GTPase (neurons) (Supplementary Fig. 15c) and Ras (intestine) (Supplementary Fig. 15d). In addition to indicating that hormetic heat stress can promote proteostasis in C. elegans, as previously shown in yeast models[52], these results also emphasize the importance of functional autophagy for maintaining proteostasis in multiple tissues, as previously suggested[53].

Finally, as the effects of HSF-1 and hormetic heat shock on stress resistance and longevity were equally dependent on autophagy, we examined whether autophagy genes were similarly required for proteostasis in animals overexpressing HSF-1. For this, we examined wild-type and HSF-1-overexpressing animals that expressed Q44::YFP in the intestine[50]. We found that the abundance of PolyQ44 aggregates was significantly lower in HSF-1-overexpressing animals than in wild-type animals (Fig. 4a,d), as previously observed in the muscle PolyQ model[3]. Moreover, RNAi of unc-51/ATG1 and bec-1/ATG6 abolished the protective effect of HSF-1 overexpression on aggregation formation, similar to our observations with hormetic heat shock (Fig. 4a,d and Supplementary Fig. 12). Therefore, we conclude that autophagy genes are also required for the proteostatic effect of HSF-1 overexpression. Collectively, the results presented here indicate that the cellular recycling mechanism of autophagy is required for the beneficial effects of hormetic heat stress and of HSF-1 overexpression on stress resistance, longevity and proteostasis.

## Discussion

In this study, we show that mild heat stress early in the life of C. elegans systemically regulates autophagy, which is essential for several health benefits conferred by hormesis, including stress resistance, lifespan extension and proteostasis. Our

heat shock regimen, which appeared to selectively induce the HSR, increased the abundance of autophagosomes in all tissues examined, probably reflecting an induction of autophagy. Interestingly, we found that heat shock increased autophagosome numbers with different kinetics in each of the examined tissues, which could be due to a number of reasons. For hypodermal seam cells, the intestine and the muscle, the endogenous lgg-1 promoter was used to drive the expression of autophagosomal marker GFP::LGG-1/Atg8, whereas the neuronal rgef-1 promoter was used for expression in the nerve ring; this difference could contribute to the different kinetics of autophagy induction in the neurons. It is also possible that each tissue perceives temperature in distinct ways via different temperature sensors. In addition, the accumulation of distinct damage in each tissue or the requirement of inter-tissue signalling for autophagy induction could be responsible for the different kinetics of autophagy induction.

Consistent with the increase in the GFP::LGG-1/Atg-8 autophagy marker, hormetic heat shock robustly increased the transcription of many autophagy genes. Although we cannot rule out an effect of heat shock on mRNA stability, transcriptional regulation has previously been implicated in the regulation of sustained autophagy[30,54]. We found that HSF-1 overexpression was sufficient to increase the mRNA levels of autophagy-related genes and the abundance of GFP::LGG-1/Atg8-positive punctae, similar to the effects of heat shock. This is consistent with a previous study showing that HSF-1 overexpression in C. elegans increased the expression of several lysosomal proteins (VHA-13, VHA-14 and VHA-15)[55]. Our findings not only suggest that basal autophagy is increased in HSF-1-overexpressing animals, but also indicate that HSF-1 overexpression is sufficient to induce autophagy. In turn, we found that hsf-1 was required for the heat shock-mediated increase in autophagosome numbers in multiple tissues. Although these findings are consistent with HSF-1 regulating autophagy in C. elegans, we note that contradictory observations on HSF-1's role in autophagy regulation have been made in other systems; lower LC3/Atg8 levels have been detected in HSF1$^{-/-}$ mice[56], whereas recent studies in human cancer cell lines show increased LC3 lipidation upon HSF1 deletion and overexpression of HSF1 prevented LC3 lipidation upon heat shock[57]. Collectively, these findings highlight the necessity for further studies to fully explore how HSF-1 may affect autophagy in specific contexts.

As two-thirds of the C. elegans autophagy-related genes examined contain at least one putative HSE in their promoter regions, it is possible that HSF-1 directly binds to the promoters to regulate autophagy gene transcription, as has previously been shown for ATG7 in breast cancer cell lines treated with the chemotherapeutic agent carboplatin[58]. Another possibility is that HSF-1 targets, such as HSPs, could regulate autophagy, as overexpression of HSP70 has been shown to inhibit starvation- or rapamycin-induced autophagy in cancer cell lines[57]. Alternatively, other stress-responsive transcription factors could play roles in inducing autophagy genes on heat shock. Consistent with this notion, we found HLH-30/TFEB to rapidly translocate into the nuclei on heat shock and hlh-30 was required for inducing the expression of several autophagy genes on heat shock. The precise contribution of HLH-30/TFEB to heat shock-mediated autophagy and its possible interaction with HSF-1 in autophagy regulation await further investigation. It will also be interesting to explore the role of known upstream regulators of autophagy, such as mTOR (mechanistic target of rapamycin), in the C. elegans response to heat stress.

Although autophagy is well recognized as a stress-inducible cytoprotective pathway, its contribution to combating stress

and promoting survival has not been well characterized. Hormetic heat shock[59] and overexpression of HSF-1 (ref. 3) are known to confer thermotolerance in *C. elegans*, possibly through enhanced capacity to cope with the damage that is caused by the elevated temperature. We found that several autophagy genes are required for the increased stress resistance of heat-shocked or HSF-1-overexpressing animals and *hlh-30* was required for the increased thermotolerance of heat-shocked animals. Previously, *daf-18*, the *C. elegans* homologue of the tumour suppressor phosphatase and tensin homologue and the gene encoding troponin-like calcium binding protein *pat-10* were the only reported effectors of increased thermotolerance conferred by hormesis[27] and HSF-1 overexpression[55], respectively. Further experiments are needed to better understand how autophagy contributes to stress resistance during hormesis and HSF-1 overexpression, and how these effector genes influence each other in the context of stress resistance. We found that autophagy genes and *hlh-30* were also required for the lifespan extension induced by hormetic heat shock and HSF-1 overexpression. Autophagy genes and *hlh-30* are similarly required for the lifespan extension of several conserved longevity paradigms, including dietary restriction, reduced insulin/insulin growth factor-1 signalling and germline removal[30]. The seemingly universal requirement for autophagy genes for longevity paradigms highlights the possibility that autophagy might have a conserved role in lifespan modulation in higher organisms.

Lastly, we showed that hormetic heat stress is sufficient to prevent the aggregation of intestinal and neuronal PolyQ proteins in *C. elegans*. Age-related diseases, such as Huntington disease, have been shown to be accompanied by autophagy dysregulation[60], and we and others have found that loss of autophagy genes abrogates the aggregation of metastable proteins and PolyQ-containing proteins[53,61]. The mechanisms by which autophagy limits PolyQ aggregation remain to be elucidated. One possibility is that increased sequestration of soluble PolyQ proteins limits their aggregation instead of possibly converting aggregates back to a soluble state. Biochemical analyses of the state of PolyQ aggregates are needed to address this question. Another possibility could be that aggregated PolyQ proteins are turned over by autophagic degradation. Therefore, it will be of interest to identify the cargo of autophagic turnover on heat stress and in PolyQ-expressing animals. The autophagy-dependent rescue of PolyQ aggregation on hormetic heat shock is particularly interesting, as this could have therapeutic implications for the treatment or prevention of diseases caused by PolyQ expansions.

In conclusion, our study demonstrates that hormetic heat shock and HSF-1 overexpression induce autophagy, which promotes the healthspan of *C. elegans*. As speculated previously[32], we propose that the interplay between stress-inducible processes, such as the HSR and autophagy, may increase the organism's ability to cope with stress (for example, thermal and proteotoxic) and ageing. As HSF-1 plays an important role in many age-related diseases in which autophagy is often deregulated, our findings suggest several therapeutic approaches for such autophagy-related diseases.

## Methods

**Strains.** Strains were maintained and cultured under standard conditions at 15 °C (for GFP::LGG-1/Atg8 punctae experiments) and 20 °C (for all other experiments) using *Escherichia coli* OP50 as a food source[62]. For RNAi experiments, animals were grown on HT115 bacteria from the time of RNAi initiation (see below). See Supplementary Table 14 for strains used and created for this study.

**RNA interference.** Gene inactivation was achieved by feeding *C. elegans* with RNAi bacterial clones expressing double-stranded RNA (dsRNA) targeting the gene of interest. Clones were obtained from the Ahringer RNAi library[63] (*atg-7*, *atg-13/epg-1*, *hlh-30*, *hsf-1*, *lgg-1/ATG8* and *wdr-23*) or the Vidal RNAi library[64] (*unc-51/ATG1*, *atg-18*, *bec-1/ATG6*, *hsp-3* and *lmp-1/LAMP1*). The *daf-2* and *isp-1* RNAi clones were previously published[65,66]. All RNAi clones were verified by sequencing.

For RNAi experiments, HT115 bacteria were grown in Luria-Bertani (LB) liquid culture medium containing 0.1 mg ml$^{-1}$ carbenicillin (Carb; BioPioneer) and 80 µl aliquots of bacterial suspension were spotted onto 6 cm nematode growth medium (NGM)/Carb plates. Bacteria were allowed to grow for 1–2 days. For induction of dsRNA expression, 80 µl of a solution containing 0.1 M isopropyl-β-D-thiogalactoside (Promega) and 50 µg ml$^{-1}$ Carb was placed directly onto the lawn. For whole-life RNAi, animals were synchronized by hypochlorous acid treatment or eggs were manually transferred onto NGM plates seeded with dsRNA-expressing HT115 bacteria. For adult-only RNAi, animals were synchronized by hypochlorous acid treatment and eggs were allowed to hatch on NGM plates seeded with OP50 bacteria. On day 1 of adulthood, animals were transferred to NGM/Carb plates seeded with dsRNA-expressing or control bacteria.

**Autophagy measurements.** Autophagy was monitored by counting GFP-positive LGG-1/Atg8 punctae in the hypodermal seam cells, body-wall muscle and proximal intestinal cells of strain DA2123 (*lgg-1p::gfp::lgg-1 + rol-6*)[33] and of strain RD202 (*unc-119; lgg-1p::gfp::lgg-1(G116A); unc-119(+)*)[35], and in the nerve-ring neurons of strain MAH242 (*rgef-1p::gfp::lgg-1 + unc-122p::rfp*)[34]. Animals were raised at 15 °C and subjected to heat shock for 1 h at 36 °C. For HSF-1-overexpressing animals, punctae were counted in the hypodermal seam cells, body-wall muscle cells and proximal intestinal cells of wild-type strain MAH236 (*lgg-1p::gfp::lgg-1 + odr-1p::rfp*)[41] and MAH534 (*lgg-1p::gfp::lgg-1 + odr-1p::rfp; let-858p::hsf-1 + rol-6*) strains, and in the nerve-ring neurons of MAH242 (*rgef-1p::gfp::lgg-1 + unc-122p::rfp*) and MAH552 (*rgef-1p::gfp::lgg-1 + unc-122p::rfp; let-858p::hsf-1 + rol-6*) strains. For RNAi experiments, animals were raised on control bacteria (empty vector) or bacteria expressing dsRNA targeting the autophagy genes *unc-51/ATG1*, *atg-18*, *bec-1/ATG6* and *lgg-1/ATG8*.

For punctae quantification, animals were mounted on a 2% agarose pad in M9 medium containing 0.1% NaN$_3$ and GFP::LGG-1/Atg8 punctae were counted using a Zeiss Imager Z1. The total number of GFP::LGG-1/Atg8 punctae was counted in all visible hypodermal seam cells, the striated body-wall muscle, the three to four most proximal intestinal cells or the nerve ring neurons. For each tissue, the total number of GFP::LGG-1/Atg8 punctae from 10 to 20 animals was counted. The average and s.e.m. were calculated and data were analysed using Student's *t*-test, one-way analysis of variance (ANOVA) or two-way ANOVA as applicable (GraphPad Prism). Data from all experiments are summarized in Supplementary Tables 3–7.

For imaging, animals were mounted on a 2% agarose pad in M9 medium containing 0.1% NaN$_3$ and images were acquired using an LSM Zeiss 710 scanning confocal microscope at × 630 magnification. GFP excitation/emission wavelengths were set at 493/523 nm to eliminate background fluorescence. For imaging of the nerve ring, Z-stack images were acquired at 0.6 µm slice intervals using an LSM Zeiss 710 scanning confocal microscope at × 630 magnification.

For bafilomycin A (BafA) experiments (that is, autophagic 'flux' assays), GFP::LGG-1/Atg8 punctae were counted in wild-type animals MAH215 (*lgg-1p::mcherry::gfp::lgg-1 + unc-122p::rfp*) and MAH236 (*lgg-1p::gfp::lgg-1 + odr-1p::rfp*)[41] and HSF-1-overexpressing animals MAH534 (*lgg-1p::gfp::lgg-1 + odr-1p::rfp; let-858p::hsf-1 + rol-6*) maintained under control conditions or subjected to 1 h heat shock at 36 °C and then injected with BafA (BioViotica) or vehicle (dimethylsulfoxide, DMSO) as previously described in ref. 37. Briefly, BafA was resuspended in DMSO to a stock concentration of 25 mM and aliquots were mixed with Blue Dextran 3000 MW (Molecular Probes) to a final concentration of 50 µM BafA in 0.2% DMSO. BafA or DMSO was injected into the anterior intestinal area and animals were allowed to recover on NGM plates with OP50 for 2 h. Surviving animals that scored positive for the blue dye were mounted on a 2% agarose pad in M9 medium containing 0.1% NaN$_3$ and imaged using an LSM Zeiss 710 scanning confocal microscope at × 630 magnification. Z-stack images were acquired at 0.6 µm slice intervals. GFP excitation/emission wavelengths were set at 493/523 nm to eliminate background fluorescence. At least 14 animals were imaged for each condition and results were combined. For hypodermal seam cells, total number of GFP::LGG-1/Atg8 punctae per seam cell were counted, and for the intestinal cells, total number of punctae per proximal cell (with visible nucleus) per 0.6 µm slice were counted. The effects of BafA could not be examined in muscle or nerve cells due to the transience of the autophagy response in these cells. The pooled average and s.e.m. were calculated and the data were analysed using two-way ANOVA (GraphPad Prism).

**Quantitative reverse tanscriptase–PCR.** Quantitative reverse tanscriptase–PCR was performed as previously described[41,67]. Briefly, total RNA was isolated from a synchronized population of ~2,000 one-day-old nematodes raised on OP50 bacteria or subjected to whole-life RNAi treatment on 6 cm NGM plates and maintained under control conditions or subjected to heat shock for 1 h at 36 °C.

For quantitative PCR analyses of older animals, the synchronized animals were washed off daily with M9 medium, adult animals were sedimented by gravity and the floating larvae were aspirated. This washing step was repeated until no more floating larvae were detected. The adult animals were re-seeded onto 10 cm NGM plates with OP50 bacteria or harvested on the desired day of adulthood. After harvesting, the animals were flash frozen in liquid nitrogen. RNA was extracted with TRIzol (Life Technologies), purified using a Qiagen RNeasy kit, and subjected to an additional DNA digestion step (Qiagen DNase I kit). Reverse transcription (1 µg RNA per sample) was performed using M-MuLV reverse transcriptase and random 9-mer primers (New England Biolabs)[68].

Quantitative reverse tanscriptase–PCR was performed using SYBR Green Master Mix in an LC480 LightCycler (Roche). A standard curve was obtained for each primer set by serially diluting a mixture of different complementary DNAs and the standard curves were used to convert the observed CT values to relative values. Three to six biological samples were analysed, each with three technical replicates. The average and s.e.m. were calculated for each mRNA. mRNA levels of target genes were normalized to the mean of the following housekeeping genes: ama-1 (large subunit of RNA polymerase II), nhr-23 (nuclear hormone receptor), cdc-42 (Rho-GTPase) and pmp-3 (putative ABC transporter)[41,69]. Housekeeping genes cdc-42 and pmp-3 were used when the data were normalized to only two housekeeping genes. Primer sequences are listed in Supplementary Table 15. Data are displayed as relative values compared with controls. Data were analysed using multiple t-test or one-way ANOVA (GraphPad Prism).

**Thermorecovery assays.** For each strain, four 6 cm NGM plates with ~20–40 animals per plate of the age indicated in Supplementary Tables 1, 9 and 11 were incubated in a single layer in a HERAtherm incubator (ThermoFisher) at 36 °C for 6–9 h. Animal survival was measured after 6–9 h at 36 °C followed by a 'recovery' period of ~20 h at 20 °C[67]. Survival was assessed by scoring the animal's voluntary movement. The average percentage survival and s.e.m. were calculated and the data were analysed by one-way or two-way ANOVA as applicable (GraphPad Prism).

**Lifespan analysis.** Lifespan was measured at 20 °C as previously described[70], using six 6 cm NGM plates seeded with OP50 bacteria or dsRNA-expressing bacteria with ~15–20 animals per plate. The L4 larval stage was recorded as day 0 of the lifespan and animals were transferred every other day to new 6 cm NGM plates throughout the reproductive period. For RNAi experiments, feeding with dsRNA-expressing or control bacteria was initiated on day 1 of adulthood. For hormetic heat shock, animals were incubated at 36 °C for 30–60 min on day 1 of adulthood. Animals were scored as dead if they failed to respond to gentle prodding with a platinum-wire pick. Censoring occurred if animals desiccated on the edge of the plate, escaped, ruptured or suffered from internal hatching. Statistical analysis was performed using Stata software (StataCorp). P-values were calculated with the log-rank (Mantel–Cox) method. See Supplementary Tables 2, 10, 12 and 13 for a summary of all lifespan experiments.

**Imaging of fluorescent reporter strains.** Whole-body fluorescence intensity was measured on day 1 of adulthood in strains BC13209 (dpy-5; atg-18p::gfp + dpy-5( + )), CF1553 (sod-3p::gfp), CL2166 (gst-4p::gfp::NLS), HZ1330 (atg-16.2p::gfp + unc-76), LD1171 (gcs-1p::gfp + rol-6), MAH325 (dpy-5; sqst-1p::gfp + dpy-5( + )), SJ4005 (hsp-4p::gfp), SJ4100 (hsp-6p::gfp) and TJ375 (hsp-16.2p::gfp) raised at 15–20 °C and then maintained under control conditions or subjected to heat shock for 1 h at 36 °C followed by 2 h recovery.

Animals were imaged on NGM plates after anaesthetization with M9 medium containing 0.1% NaN3. Images were acquired with a Leica DFC310 FX camera using the exposure times indicated in the figure legends. Image analysis was performed with ImageJ software (National Institutes of Health) by tracing the individual animals and measuring the mean intensity of GFP fluorescence per animal.

Nuclear localization of HLH-30 was imaged on day 1 of adulthood in strain MAH235 (hlh-30p::hlh-30::gfp + rol-6) raised at 20 °C and then maintained under control conditions or subjected to heat shock for 1 h at 36 °C. Animals were imaged using a Zeiss Imager Z1 at × 100 magnification or with a Leica DFC310 FX camera.

**Pharyngeal pumping and food deprivation assays.** Pharyngeal pumping was measured on day 1 of adulthood in wild-type animals raised at 20 °C and then maintained under control conditions or subjected to heat shock for 15–60 min at 36 °C followed by 2 h recovery. Pharyngeal pumping was measured by counting the grinder movements in the terminal bulb of 16–20 animals for 15 s on a Leica stereoscope.

Short-term food deprivation was performed in wild-type animals raised at 20 °C. On day 1 of adulthood, animals were washed off with liquid M9 media and, after several washing steps, animals were either kept in liquid M9 or seeded in small aliquots (20 worms) on 6 cm plates containing 2% agarose in ddH2O. Animals were kept without food for 90 min and then either seeded or transferred to 6 cm NGM plates containing OP50 bacteria as a food source.

**Analysis of PolyQ strains.** The number of intestinal PolyQ aggregates was counted in individual animals of strains GF80 (Ex(vha-6p::Q44::yfp + rol-6)) and MAH602 (Is(vha-6p::Q44::yfp + rol-6)) on day 2–5 of adulthood. Animals were raised at 20 °C and then maintained under control conditions or subjected to heat shock for 1 h at 36 °C on day 1 of adulthood. HSF-1-overexpressing animals MAH575 (hsf-1p::hsf-1::gfp + rol-6; Ex(vha-6p::Q44::yfp + rol-6)) were also analysed. Animals were imaged on NGM plates without food after anaesthetization with M9 medium containing 0.1% NaN3. Images were acquired with a Leica DFC310 FX camera. Aggregates were counted using the 'Cell counter' function of ImageJ software.

The number of neuronal PolyQ aggregates was counted in the nerve ring neurons of individual animals of strain AM101 (rgef-1p::Q40::yfp) on day 5–9 of adulthood. Animals were raised at 20 °C and then maintained under control conditions or subjected to heat shock for 1 h at 36 °C on day 1 of adulthood. Animals were imaged on NGM plates or microscope slides after anaesthetization with M9 medium containing 0.1% NaN3 using either a Leica DFC310 FX camera, or a Zeiss Imager Z1 at × 100 magnification. Z-stack images were acquired at 0.6 µm slice intervals using an LSM Zeiss 710 scanning confocal microscope at × 630 magnification.

**Data availability.** The authors declare that all data supporting the findings of this study are available within this article, its Supplementary Information files, the peer-review file, or are available from the corresponding author upon reasonable request.

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

## Acknowledgements

We thank Drs Ao-lin Hsu, Veena Prahlad, and Jeffery W. Kelly for input on the project; Dr Chung-Yi Liang, Dr Sara Gelino, Andrew Davis and Linnea Adams for technical assistance; Dr Anne O'Rourke for comments on the manuscript; and Drs Jian-Liang Li and Alexey Eroshkin for help with bioinformatics. We thank Drs Renaud Legouis, Rick Morimoto, Hong Zhang, and Tali Gidalevitz for kindly providing strains. Some of the nematode strains used in this work were provided by the *Caenorhabditis* Genetics Center (University of Minnesota), which is supported by the NIH–Office of Research Infrastructure Program (P40 OD010440). C.K. was supported by a postdoctoral fellowship from American Federation for Aging Research (AFAR) (EPD1360) and M.H. was supported by NIH/NIA grants R01 AG038664 and R01 AG039756, and by a Julie Martin Mid-Career Award in Aging Research supported by The Ellison Medical Foundation and AFAR..

## Author contributions

C.K. and M.H. designed the experiments. J.T.C. performed part of the BafA injections (Figs 1 and 2). J.S. imaged the heat stress reporter strain after autophagy RNAi (Supplementary Fig. 4a) and performed repeats of the behavioral analysis of the temperature-sensitive misfolding mutants after autophagy RNAi (Supplementary Fig. 15). C.K. performed all other experiments and analysed the data. C.K. and M.H. wrote the manuscript.

## Additional information

**Competing financial interests:** The authors declare no competing financial interests.

**Publisher's note**: 

