## [Peer Review File · Nature Communications]

Reviewer #1 (Remarks to the Author)

In this manuscript, Kumsta and colleagues investigate the interplay between two stress response pathways, the heat shock response (HSR) and autophagy, using *C. elegans* as a model system. They show that heat stress induces the expression of heat shock response genes that are transcriptionally regulated by the heat shock transcription factor -1 (HSF -1) as well as the expression of several autophagy genes. In addition, they find that autophagy is induced after heat shock in a tissue- specific manner. Interestingly, they show that autophagy genes are required for the enhanced stress resistance and longevity conferred by hormetic heat stress and HSF-1 overexpression. Along similar lines, exposure to mild heat stress at day 1 of adulthood reportedly protects against polyQ toxicity in a nematode model of Huntington disease. Again, autophagy appears to be required for the proteostatic benefits of hormetic heat shock. Together, these findings establish a common axis and a cooperation of HSR and autophagy as a mechanism to cope with stressful conditions. The study suggests that autophagy is required for HSF-1 -regulated functions in the HSR, protein homeostasis and ageing. However, the concept of the interplay between the heat shock response and autophagy is hardly new. This area of research has attracted much attention in recent years and there is a wealth of reports in the literature describing the relationship between these two major homeostatic systems. In addition, it has already been demonstrated that heat shock induces autophagy in a tissue-specific manner (Alavez et al, 2011; Dokladny et al, 2013; Desai et al, 2013; Tonq et al, 2014; Dokladny et al, 2014; reviewed by Dokladny et al, 2015). Overall, this manuscript does not significantly add to this prior art. More importantly, the study falls short of providing any real mechanistic insight as to how the interaction between the heat shock response and autophagy increases thermotolerance and protects against proteotoxicity.

Specific Comments

The authors should check the expression levels of autophagy in HSF-1 deficient animals under normal and heat stress conditions to fully justify that autophagy genes are transcriptionally regulated from HSF-1 as part of the HSR.

In Figure 1A, genes related to autophagy, fusion and H⁺ pumps appear more increased when HSF-1 is overexpressed than when animals are heat-shocked and HSF-1 not overexpressed. The case is opposite compared to the heat shock proteins. How do the authors explain the fact OE of HSF-1 doesn't increase the Heat Shock genes upon Heat Shock, as it does for autophagy related genes?

The authors cannot be sure that autophagy deficiency does not compromise the organism's ability to induce the HSR just by monitoring hsp-12.6 expression levels when autophagy is inactive, because hsp-12.6 is not a sole HSF-1 target as shown by Honjoh et al., 2009, Nature. 457: 726. Therefore, additional targets together with the HSF-1 localization and granulation should be shown under normal and heat stress conditions of defective autophagy.

Is the differential tissue-specific response to heat stress due to cell non-autonomous effects? Additional heat stress conditions should be checked before general conclusion about tissue-specific responses to heat stress can be drawn (i.e. other temperatures). Is thermotolerance of worms that don't overexpress HSF-1 reduced upon autophagy knock-down?

What happens if hormetic heat shock is induced after the reproductive period, at day 4? Or just in day 2 or 3 of adulthood? Does it still have a protective effect? Animals subjected to hormetic heat stress were more resistant to thermal stress. What about resistance of these animals to other types of stresses as well?

Is the induction of autophagy genes upon heat stress mediated directly by HSF-1 at the transcriptional level? Is it indirect? While the fact that overexpression of HSF-1 is sufficient to boost the expression of autophagy genes points to that direction, this should be further verified by the reverse experiment, by using hsf-1 mutants or RNAi.

While hsp-16.2 expression can be efficiently induced following heat shock (Sup. Figure 2E), so autophagy is not required for HSR induction, animals subjected to heat shock following a hormetic shock are more susceptible. Can this solely be explained by protein aggregation phenomena in autophagy-deficient animals?

The authors find that autophagy genes were similarly upregulated in heat stressed animals and in animals over-expressing HSF-1 under non-stressed conditions, concluding that HSF-1 plays an important role in the transcriptional regulation of autophagy genes upon stress. To confirm their statement they should show that autophagy genes are not upregulated in nematodes lacking the hsf-1 gene. The possibility exists that an independent posttranscriptional mechanism can regulate the abundance of various transcripts, such as autophagy gene transcripts.

Given that both chaperone function and autophagy activity decline with age, the authors should also examine nematodes older than the 5-day-old adults in order to test the importance of autophagy genes in proteostasis conferred by hormetic heat stress and HSF-1 during ageing.

Besides HSF-1, the authors could test the requirement for additional stress response transcription factors such as SKN-1, which is known to promote proteostasis through the control of genes encoding chaperones and autophagy proteins, among others.

It is known that autophagy inhibition in wt worms limits their lifespan as it is also obvious in some of the curves. The fact that the hormetic effect of heat stress on lifespan is not achieved when autophagy is inhibited could be a result of a general, non-specific detrimental effect of autophagy inhibition on organismal physiology rather than specific inhibition of the hormetic effect.

Minor points

Hormesis refers to the beneficial effects of a mild stressor or treatment that becomes deleterious at higher levels (Gems and Partridge, 2008). The definition given (on page 7) rather refers to antagonistic pleiotropy (Williams, 1957).

Why were HSF-1-overexpressing animals fed bacteria expressing dsRNA against autophagy genes from hatching, while wild-type animals were subjected to RNAi from day 1 of adulthood? (legend of Fig. 2). As the authors report (pages 8 and 9) and Figure 3 shows, hormetic heat shock on day 1 of adulthood caused a significant reduction in the level of intestinal PolyQ44 aggregates on days 2-5. However, according to information provided in the Methods section, it is not clear if the number of aggregates or the mean fluorescence intensity of GFP signal per animal was measured.

Figure S3: I would recommend that the authors explain more explicitly in the text the bafilomycin experiment and how it proves the fact that the heat-shock induced LGG-1 puncta are due to induction of autophagy and not due to inhibition of autosome turnover. I believe that the samples that need to be directly compared also in the bar graph are number 2 and 3 and number 2 and 4.

Figure 2B: The title above the graph could change to something more descriptive and not repetitive of the x axis, like "thermotolerance".

Authors should use control RNAi in their experiments, to exclude putative nonspecific RNA silencing effects.

Key experiments should be repeated with stable genetic mutants instead of with RNAi lines. If possible, constitutively active HSF-1 mutants should be examined in addition to HSF-1 overexpressing animals.

In Figure 1B-D authors should add representative images of the HSF-1 overexpressing sample as

well.

In figure 3A a positive control (autophagy induction condition) is needed.

The last sentence of the abstract is syntactically incorrect.

On page 10, second paragraph, first sentence, I believe it's missing 1 word: ... HSF-1 was sufficient to increase the expression of autophagy-related genes....

Reviewer #2 (Remarks to the Author)

Kumsta et al., (Hansen) investigated the interplay between the heat-shock response and autophagy in *C. elegans*. The authors show that a reporter of autophagy and certain autophagy (Atg) genes are induced following heat-shock and in animals overexpressing HSF-1, and that autophagy genes are required for the beneficial effects on stress resistance and lifespan conferred by both HSF-1 overexpression and hormetic heat stress. They also show that hormetic stress reduces the aggregation of polyglutamine in intestinal cells in an autophagy dependent manner.

While these observations could be of broad interest to the field, the results are incomplete and will require additional clarification. Perhaps some of this problem is because the paper has two components on the heat shock effects on autophagy and the possible involvement of HSF-1, and somewhat separately the observations on hormesis.

1) The central observation of this paper is that different reporters of autophagy are induced in response to heat shock. While not questioning the statistical significance of induction of the Atg genes upon heat shock, it is notable that the level of induction is quite small relative to the reference heat shock genes that show substantial induction. This brings forth many questions, for example whether this effect is a transcriptional or post-transcriptional response as heat shock also can affect mRNA stability. Of course, the latter observation may be inconsistent with the requirement for HSF-1. Despite the evidence that HSF-1 is involved, it is not possible to conclude from these experiments whether this is a primary consequence of HSF-1 binding to heat shock elements in the promoters of the Atg genes, a secondary consequence of the activity of an HSF-1 directly regulated gene, or that the Atg effect is really HSF-1 independent but heat shock dependent. The effects of heat shock on autophagy shows that GFP::LGG-1/Atg8 forms puncta, however it is unclear how this relates to changes in autophagic activity. Further, how does the induction of a specific subset of Atg genes by heat shock result in increased macroautophagy in a stressed cell? Are these Atg genes that are induced important regulatory components or is it proposed that the entire macroautophagy machinery is upregulated?

2) In looking at Figure 1, the HSF-1 dependence of Atg gene induction could be directly tested either using HSF-1 mutations or RNAi. Could heat-shock or overexpression of HSF-1 affect feeding behavior and thus affect the expression of autophagy genes independently of HSF-1? While direct ChIP experiments could be done, it would also be useful to search different datasets of HSF-1 chromatin binding in different organisms to see whether there is a direct regulation of Atg genes and if this is conserved? While answers to these questions would require substantial additional experiments, some of this can also be determined by standard genomics informatics and by generating transgenic lines expressing Atg heat shock inducible gene promoter reporter constructs. Such observations would increase an enthusiasm to embark on the chromatin experiments to demonstrate whether HSF-1 binds directly to Atg gene promoters. There is also the alternative possibility that HLH-30 has a role in heat-shock induction of Atg genes.

3) The observations in Figures 1 and S2, the observations on that GFP::LGG-1/Atg8 puncta increase upon heat shock and in Atg mutants? Often puncta of soluble proteins in cells exposed to heat shock is interpreted as protein aggregates. In Figure S2, why does RNAi of Atg genes have little or no effect on foci numbers in the control?

4) While the use of Bafilomycin A treatment, by direct injection into animals, can be interpreted that the observed increase in Igg-1 foci is due to the increased generation of autophagosomes, further experiments are necessary to characterize autophagy flux during heat-shock and the role of HSF-1.

5) Some of the data on tissue-specific analyses of autophagy are confusing. For example, the data presented in Table S1 are not consistent between the tissues (for example only day 1 was analyzed for muscle), certain time points presented in Fig. S4 were analyzed only once according to Table S1 or are even absent from the Table, and in Fig. S4 the scale representing the experiment is not clear. Is the time point 0h before the HS or after? This should be clarified. The authors draw conclusions about tissue-specific dynamics of autophagy induction upon heat-shock, however, some of these results could also come from the experimental design. For example, the promoters used to drive the expression of the Igg-1::GFP protein fusion could result in differential tissue expression (for example in neurons), rendering the results difficult to compare. Likewise, could the different kinetics observed be due to differential tissue sensitivity to the effects of heat shock?

6) It is shown that Atg genes are required for the beneficial effect of hormetic stress on survival to heat shock at day 4 of adulthood. However, in Table S5, Atg gene RNAi seems to also have an effect on the survival of the control animals at day 4. In these results what are the p-values comparing? As the conclusion of these experiments would seem more central, perhaps they should be in Figure 2. Did the authors try the hormetic stress at other days of adulthood to assess the protective effect?

7) Perhaps slightly outside the central theme of heat shock and autophagy, have the authors considered the mechanism of hormetic stress mediated stress resistance? In addition to expected roles of HSF-1, ATFS1, XBP1, ATF-6 are other stress factors required? How about HLH-30? Directly relevant to the hormesis observations, are heat shock and autophagy genes more highly induced in animals pre-treated with heat shock versus control animals upon heat-shock at D4? If so, is this due to better HSF-1 or HLH-30 binding to Atg gene promoters?

8) It is an interesting observation that a hormetic heat shock exposure prevents the aggregation of polyQ in intestine and that autophagy genes are required. What is the effect on the biochemical state of polyQ aggregates? If this is linked to autophagy one would expect to see degradation and therefore reduced levels and not a conversion to soluble protein as would occur by enhancing chaperones. Is autophagy directly involved in clearance of polyQ aggregates or is this a chaperone-dependent process of the chaperone disaggregases? Further, given that polyQ expressing animals are impaired, is autophagy itself induced in polyQ expressing animals? Finally, as polyQ models are available in other tissues, is the hormetic stress and autophagy enhancement protective in these other models?

Response to reviewers for manuscript NCOMMS-16-10042

We enclose our revised manuscript entitled “**Hormetic heat stress induces autophagy via HSF-1 to improve survival and proteostasis in *C. elegans***” by Kumsta *et al.* We appreciate the reviewers’ insightful comments, which we have addressed point-by-point below (in blue text). In response to this useful feedback, we have made the following major textual changes and experimental additions (a complete list of tables and figures can be found at the end of this document):

1. To better illustrate the **novelty and impact** of our paper, we have changed the title and abstract and restructured the Results section to better emphasize our key finding that autophagy is engaged as a protective mechanism in response to hormesis in the multi-cellular organism *C. elegans*. In particular, we characterize the effect of this beneficial heat shock on the autophagy response in a comprehensive and tissue-specific manner. Moreover, we demonstrate that a mild heat shock early in life protects against PolyQ protein aggregation in an autophagy-dependent manner, highlighting a possible disease intervention to be further explored in other systems.
2. We have investigated the effects of ***hsf-1* knockdown** on heat shock-mediated autophagy, and found that reduction of *hsf-1* by RNAi limited the induction of GFP::*LGG-1/Atg8*-punctae upon heat stress in multiple major tissues. This finding suggests that *hsf-1* is required for heat shock-induced autophagy. See Reviewer 1, point A below for a full discussion of experiments using *hsf-1(RNAi)*.
3. To address whether **HSF-1 regulates autophagy gene expression directly or indirectly**, we have analyzed the promoter sequences of autophagy genes in *C. elegans* and found heat shock elements (HSEs) in 42 out of 63 analyzed genes. Together with our collaborators in Dr. Allen Hsu’s lab, we used ChIP to test whether HSF-1 can bind to putative HSEs in the promoters of *bec-1*, *atg-18* and *sqst-1* after a 30 min heat shock in wild-type animals. In these preliminary efforts, we did not observe significant binding of HSF-1 to these promoter elements, and we therefore cannot exclude that other transcription factors play roles in regulating autophagy gene expression in response to heat shock (see no. 4 just below).
4. We have analyzed and determined a novel **role for the TFEB homolog HLH-30**, a conserved regulator of the autophagy response, in regulating heat-shock-mediated autophagy. Specifically, we show that heat shock leads to rapid nuclear translocation of HLH-30, and *hlh-30* loss-of-function mutants have limited induction of autophagy genes upon heat shock and do not benefit from a hormetic heat shock in terms of survival and longevity.
5. We have characterized the effects of heat shock on **pharyngeal pumping** and ruled out a contribution of short-term food deprivation on hormetic benefits of survival and longevity.
6. We have investigated the effects of **hormetic heat shock in *C. elegans* at older ages** and found the hormetic benefits persist through the animals’ reproductive phase in terms of survival and longevity.
7. We have expanded our analyses of the **beneficial effects of a hormetic heat shock on proteostasis**. Similar to the intestine, we have found that hormetic heat shock reduces PolyQ aggregation in neurons in an autophagy-gene-dependent fashion. Consistent with these findings, we show that autophagy genes are induced in response to heat shock in animals expressing PolyQ. Finally, we have measured the lifespan of animals expressing PolyQ-proteins upon heat shock to demonstrate that the hormetic heat shock has beneficial long-term effects.

With these novel insights, as well as several other advancements highlighted below, we believe we have greatly improved the revised manuscript.

Reviewer #1: In this manuscript, Kumsta and colleagues investigate the interplay between two stress response pathways, the heat shock response (HSR) and autophagy, using *C. elegans* as a model system. They show that heat stress induces the expression of heat shock response genes that are transcriptionally regulated by the heat shock transcription factor -1 (HSF -1) as well as the expression of several autophagy genes. In addition, they find that autophagy is induced after heat shock in a tissue- specific manner. Interestingly, they show that autophagy genes are required for the enhanced stress resistance and longevity conferred by hormetic heat stress and HSF-1 overexpression. Along similar lines, exposure to mild heat stress at day 1 of adulthood reportedly protects against polyQ toxicity in a nematode model of Huntington disease. Again, autophagy appears to be required for the proteostatic benefits of hormetic heat shock. Together, these findings establish a common axis and a cooperation of HSR and autophagy as a mechanism to cope with stressful conditions. The study suggests that autophagy is required for HSF-1 -regulated functions in the HSR, protein homeostasis and ageing. However, the concept of the interplay between the heat shock response and autophagy is hardly new. This area of research has attracted much attention in recent years and there is a wealth of reports in the literature describing the relationship between these two major homeostatic systems. In addition, it has already been demonstrated that heat shock induces autophagy in a tissue-specific manner (Alavez et al, 2011; Dokladny et al, 2013; Desai et al, 2013; Tonq et al, 2014; Dokladny et al, 2014; reviewed by Dokladny et al, 2015). Overall, this manuscript does not significantly add to this prior art. More importantly, the study falls short of providing any real mechanistic insight as to how the interaction between the heat shock response and autophagy increases thermotolerance and protects against proteotoxicity.

We have taken the reviewer's comments to heart and have changed the main focus of the manuscript and better emphasized previous discoveries on the effect of heat shock on autophagy. In the introduction on p. 3, we now state:

"While heat shock can modulate autophagy in different cell models, and it has been shown that the HSF-1-regulated HSR and autophagy can be coordinated under certain stress conditions (reviewed in³²), it remains unclear how autophagy contributes to stress resistance in organisms subjected to stressors such as a hormetic heat shock."

We do, however, respectfully disagree that our study does not add new findings to the concept of the interplay between the heat shock response and autophagy. We provide the first comprehensive analysis of the effects of heat shock on autophagy in multiple tissues of an organism; we uncovered tissue-specific differences in this response; we characterized the physiological effects of heat-shock-induced autophagy, and suggest that the two transcription factors HSF-1 and HLH-30/TFEB play regulatory roles in this response.

Specific Comments

A. The authors should check the expression levels of autophagy in HSF-1 deficient animals under normal and heat stress conditions to fully justify that autophagy genes are transcriptionally regulated from HSF-1 as part of the HSR.

We have now monitored autophagy in multiple ways in *hsf-1(RNAi)* animals (new **Figure S8**, discussed on page 8). We have found that *hsf-1(RNAi)* animals did not display increased numbers of GFP::LGG-1/Atg8-punctae upon heat stress in the three major tissues we analyzed (intestine, muscle, and neurons), which suggests that *hsf-1* is required for heat shock-induced autophagy.

We also used quantitative RT-PCR to analyze autophagy gene transcript levels in *hsf-1(RNAi)* animals under control conditions and upon heat shock (**Figure R1**). Curiously, we found that *hsf-1(RNAi)* animals have increased levels of the majority (5 out of 6) of autophagy-related genes analyzed under control conditions. This increase in autophagy-related genes could be induced as a compensatory mechanism to intrinsic protein aggregation caused by reduced HSF-1 levels. In such a

scenario, *hsf-1(RNAi)* animals may be trying to induce autophagy, but fail to fully form autophagosomes in the major tissues where an induction in GFP::*LGG-1* positive punctae upon heat shock were not seen. While we observe that autophagy genes *unc-51/ATG1*, *atg-18*, and *bec-1/ATG6* were not further induced upon heat shock, *atg-9*, *lgg-1/ATG8* and *sqst-1* were increased, albeit to a lower degree than in wild-type animals. Whether this induction is due to residual activity of *hsf-1* or whether other transcription factors (e.g., HLH-30, see below) are responsible for the induction of these genes remains unclear. Clearly, more experiments are needed to fully understand what is going on in *hsf-1(RNAi)* animals, and we would prefer not to include these data in the main manuscript. Instead, we recognize that it is currently unclear if the effects of HSF-1 in regulating autophagy either by hormetic heat shock or by HSF-1 overexpression, occur via direct or indirect means (to this point, please also see response to point H below on ChIP efforts).

B. In Figure 1A, genes related to autophagy, fusion and H⁺ pumps appear more increased when HSF-1 is overexpressed than when animals are heat-shocked and HSF-1 not overexpressed. The case is opposite compared to the heat shock proteins. How do the authors explain the fact OE of HSF-1 doesn't increase the Heat Shock genes upon Heat Shock, as it does for autophagy related genes?

We found that autophagy-related genes were 5-10-fold induced upon heat shock (**Figure 1**), and about 10-20-fold increased in HSF-1 overexpressing animals (**new Figure 2**). In our HSF-1 overexpressing strain, heat-shock protein genes were also elevated about 20-fold under non-stress conditions. However, it was previously shown that overexpression of HSF-1 does not necessarily lead to the increase of heat-shock proteins under control conditions (Baird et al, Science, 2014). We would like to clarify that we have not subjected the HSF-1-overexpressing animals to heat shock and that all experiments with the HSF-1-overexpressing animals have been performed under control conditions.

C. The authors cannot be sure that autophagy deficiency does not compromise the organism's ability to induce the HSR just by monitoring *hsp-12.6* expression levels when autophagy is inactive, because *hsp-12.6* is not a sole HSF-1 target as shown by Honjoh et al., 2009, Nature. 457: 726. Therefore, additional targets together with the HSF-1 localization and granulation should be shown under normal and heat stress conditions of defective autophagy.

We would like to clarify that we have not monitored *hsp-12.6* expression levels, but rather *hsp-16.2* expression levels. The promoter region of *hsp-16.2* has two HSEs (Fernandes et al., 1994; Trinklein et al., 2004; Guertin and Lis, 2010), and also contains a heat shock-associated site (HSAS) (GuhaThakurta et al., Genome Research, 2002). Both promoter regions are bound by HSF-1, and we are not aware that *hsp-16.2* is transcriptionally regulated by transcription factors other than HSF-1. Depletion of *hsf-1* by RNAi abolishes the expression of *hsp-16.2* upon heat shock, and we have added these images to **Figure S4A**.

To address the reviewer's concern, we have now analyzed a second heat shock-inducible reporter, namely *hsp-70p::gfp* (C12C8.1) in animals with decreased levels of autophagy genes (**new Figure S4B**). The capacity to induce a heat shock response is not diminished in these animals, rather we find that *atg-18* and *lgg-1* RNAi lead to increased expression of the *hsp-70p::gfp* reporter upon heat stress. Consistent with this observation, we also observed that HSF-1 stress body formation in response to heat stress was not affected by autophagy RNAi, (**Figure R2**). Lastly, we have analyzed the transcript levels of four heat shock protein genes in animals treated with autophagy gene RNAi and found these animals to not be compromised in their ability to induce a HSR (**new Figure S4C and Figure R3**). In the manuscript, we have only included results of animals subjected to *lgg-1* and *bec-1* RNAi, as we were able to confirm *lgg-1* knockdown and the presence of *bec-1* dsRNA in these samples. In conclusion, with these additional experiments, we have no indication that autophagy gene reduction compromises the ability of the animal to mount a HSR.

D. Is the differential tissue-specific response to heat stress due to cell non-autonomous effects? Additional heat stress conditions should be checked before general conclusion about tissue-specific responses to heat stress can be drawn (i.e. other temperatures).

We agree with the reviewer that it will be interesting to determine whether the tissue-specific responses to heat shock are due to cell-nonautonomous effects. We have now expanded our discussion on the tissue-specific differences on p. 14, where we say:

“Interestingly, our tissue-specific GFP::LGG-1/Atg8 marker analysis of the autophagy response to heat shock demonstrated that every tissue examined is distinct in their response of autophagy induction. This could be for a number of reasons. For hypodermal seam cells, the intestine and the muscle, the endogenous *lgg-1* promoter was used to drive the expression of autophagosomal marker GFP::LGG-1/Atg8, whereas the neuronal *rgef-1* promoter was used for expression in the nerve ring; this difference could contribute to the different kinetics of autophagy induction in the neurons. Moreover, it is possible that the different tissues perceive temperature in distinct ways and involve different temperature sensors. Additionally, the accumulation of distinct damage in each tissue, or the requirement of inter-tissue signaling for autophagy induction could be responsible for the differential autophagy induction.”

Instead of investigating other temperatures, we have analyzed tissue-specific responses in the absence of transcription factors HSF-1. These results indicate that individual tissues have a specific response to heat stress, which will be interesting to investigate in detail in future studies.

E. Is thermotolerance of worms that don't overexpress HSF-1 reduced upon autophagy knock-down?

Similarly to lifespan experiments, the adult-only reduction of autophagy genes does not significantly reduce the thermo-recovery of wild-type animals (see **Table S1**). To help illustrate this point, we have now added *P*-values to compare autophagy genes RNAi to empty vector controls into the table.

F. What happens if hormetic heat shock is induced after the reproductive period, at day 4? Or just in day 2 or 3 of adulthood? Does it still have a protective effect?

To address this interesting point, we have performed multiple new experiments. We have analyzed the effect of a 1-hour heat shock on animals on day 1, 3, 5, and 7 of adulthood, and have analyzed their thermo-recovery, lifespan and measured their ability to induce autophagy genes. These new data indicate that animals gain beneficial effects from a hormetic heat shock through day 5 of adulthood, but cease at older ages. Our data are in line with previous studies that the ability of animals to respond to a hormetic treatment decreases with age (day 8 or older) (Olsen et al, Biogerontology, 2006). We have included these data in **new Figure S11** and **Tables S1-2**, and discuss the new data on p. 10, where we say:

“To finally address whether a hormetic heat shock would have beneficial effects later in life, we heat-shocked animals on day 1, day 3, day 5, and day 7 of adulthood and analyzed their thermo-recovery two days later. A hormetic heat shock up to day 5 of adulthood was capable of inducing stress resistance compared to untreated animals with the greatest effect on day 1 of adulthood, and no significant effect on day 7 (**Fig. S11A and Table S1**). Similarly, lifespan analyses revealed that animals were able to consistently benefit from a hormetic heat shock up to day 5 of adulthood (**Fig. S11B and Table S2**), even though the ability of animals to respond to the hormetic treatment decreased with age, as previously reported⁴⁷. Additionally, we measured the mRNA levels of heat shock protein genes *hsp-70* and *hsp-16.1* as well as several autophagy genes (i.e., *bec-1/ATG6*, *sqst-1*, *atg-18* and *lgg-1/ATG8*) in animals that were heat shocked later in life. While heat shock protein genes and autophagy genes *bec-1/ATG6* and *sqst-1* remained heat-inducible on day 1 through 7 (with a slight time-dependent reduction in magnitude of induction), *atg-18* and *lgg-1/ATG8* were only induced by heat shock on day 1 of adulthood. These findings are consistent with the notion that autophagic activity declines with age^{48, 49}.”

G. Animals subjected to hormetic heat stress were more resistant to thermal stress. What about resistance of these animals to other types of stresses as well?

It has previously been shown that cross-tolerance can be induced with two different oxidative stressors (Cypser, et al., J Gerontol A Biol Sci Med Sci, 2002). However, we have not tested the ability of cross-tolerance induced by hormetic heat shock, since we feel it is better in line with our overall study to thoroughly characterize the heat-shock paradigm. For example, we already tested whether our heat-shock regimen could induce other stress sensors (i.e., oxidative stress, mitochondrial stress, and ER stress) and demonstrated a very limited induction of the ER stress reporter strain, which expresses *gfp* under the control of the *hsp-4* promoter (now **Figure S1A**). To further support these studies, we have now also performed quantitative RT-PCR to test whether stress response genes are up-regulated with our heat-shock regimen and in the HSF-1 overexpressing strain (**new Figure S1B**). In these experiments, we observed a limited induction of *hsp-6* (upregulated upon mitochondrial stress) and *gst-4* (upregulated upon oxidative stress), which may indicate a possible, but likely limited ability of cross-tolerance to other stressors by heat-shock treatment. Future experiments will be needed to investigate this point in detail. In the manuscript, we describe these results on p. 5 of the manuscript, where we say:

“This heat shock selectively induced the heat shock response with an induction of heat shock protein genes such as *hsp-70* and *hsp-16.2* and only mildly increased the expression of the mitochondrial stress gene *hsp-6* and oxidative stress gene *gst-4*, but not other markers of oxidative or endoplasmic reticulum stress responses (**Fig. S1**).”

H. Is the induction of autophagy genes upon heat stress mediated directly by HSF-1 at the transcriptional level? Is it indirect? While the fact that overexpression of HSF-1 is sufficient to boost the expression of autophagy genes points to that direction, this should be further verified by the reverse experiment, by using *hsf-1* mutants or RNAi.

We agree with the reviewer that it would be interesting to determine whether the regulation of HSF-1 is direct or indirect. To this end, we have analyzed the promoter sequences (2000 bp upstream of the start site) of autophagy genes in *C. elegans* and found putative HSEs for 42/63 autophagy-related genes. In addition, together with our collaborators in Dr. Allen Hsu's lab, we have designed primers spanning putative HSEs in the promoters of select autophagy genes and performed preliminary ChIP analyses on whole animal extracts from wild-type animals that were heat shocked for 30 min at 35°C (this protocol, using a novel HSF-1 antibody, is still under development in Dr. Hsu's lab). Under these conditions, we did not find HSF-1 bound to these putative HSEs in promoter regions of autophagy genes *atg-18*, *bec-1/ATG6* and *sqst-1* (“**REDACTED**”). There could be a number of reasons for this negative observation, including that the heat paradigms may differ slightly (performed in different labs). Without more experimental effort, we can therefore not exclude that the induction of autophagy by HSF-1 occurs through indirect means. Since we firmly believe that this important point is best addressed in a future in-depth analysis using ChIP-seq, we have instead clearly noted that the mechanism of action of HSF-1 is currently unclear, and could include either direct or indirect mechanisms. To this end, on p. 14-15, we say:

“We conclude that HSF-1 is a novel regulator of autophagy in *C. elegans*, and it will be of great interest to determine the exact mechanism by which HSF-1 exerts its effects on autophagy. Since two thirds of the *C. elegans* autophagy-related genes we examined contained one or more putative heat shock elements (HSEs) in their promoter regions, it is possible that HSF-1 could directly bind to autophagy gene promoters and regulate autophagy gene transcription. This has previously been shown for *ATG7* in breast cancer cell lines upon treatment with the chemotherapeutic agent carboplatin⁵⁶. Another possibility is that HSF-1 targets, such as heat shock proteins, could regulate autophagy, since overexpression of *HSP70* has been shown to inhibit starvation- or rapamycin induced autophagy in cancer cell lines⁵⁷.”

Please see response to point A above for experiments analyzing *hsf-1(RNAi)* animals.

I. While *hsp-16.2* expression can be efficiently induced following heat shock (Sup. Figure 2E), so autophagy is not required for HSR induction, animals subjected to heat shock following a hormetic shock are more susceptible. Can this solely be explained by protein aggregation phenomena in autophagy-deficient animals?

To clarify, wild-type animals with reduced levels of autophagy genes are able to mount a heat-shock response and are not compromised in their thermo-tolerance or lifespan (**Table S1-2**). Animals with reduced levels of autophagy are however not able to benefit from a hormetic heat stress.

J. The authors find that autophagy genes were similarly upregulated in heat stressed animals and in animals over-expressing HSF-1 under non- stressed conditions, concluding that HSF-1 plays an important role in the transcriptional regulation of autophagy genes upon stress. To confirm their statement they should show that autophagy genes are not upregulated in nematodes lacking the *hsf-1* gene.

Please see response to point A above.

K. The possibility exists that an independent posttranscriptional mechanism can regulate the abundance of various transcripts, such as autophagy gene transcripts.

We agree with the reviewer that independent posttranscriptional mechanisms could lead to autophagy induction upon heat shock and we have now discussed this on p. 15 of the manuscript, where we say:

“It will also be interesting to explore the role of known upstream regulators of autophagy, such as the mechanistic target of rapamycin (MTOR) in the response to heat stress in *C. elegans*.”

L. Given that both chaperone function and autophagy activity decline with age, the authors should also examine nematodes older than the 5- day-old adults in order to test the importance of autophagy genes in proteostasis conferred by hormetic heat stress and HSF-1 during ageing.

As also noted in point F, we have now examined whether a hormetic heat shock is beneficial in animals at older ages, and we found that animals no longer benefit from a hormetic heat shock on day 7 of adulthood. Interestingly, we observed that these older animals still had elevated expression of *hsp-70*, *hsp-16.1* (albeit to a lesser degree than on day 1) upon heat shock. Additionally, the transcript levels of *bec-1/ATG6* were induced to similar levels on later days as on day 1, whereas no induction was observed of *sqst-1*, *atg-18* or *lgg-1/ATG8*. Further experiments will be necessary to precisely determine why older animals are not able to benefit from hormetic heat stress.

M. Besides HSF-1, the authors could test the requirement for additional stress response transcription factors such as SKN-1, which is known to promote proteostasis through the control of genes encoding chaperones and autophagy proteins, among others.

In our initial analysis of whether our hormetic heat shock regimen could induce stress response genes other than those involving heat shock genes, we included the *gcs-1p::gfp* reporter (*gcs-1* is a SKN-1 target gene) as well as *gst-4p::gfp* reporter (*gst-4* is both a SKN-1 and DAF-16 target gene). For our resubmission, we measured the transcript levels using quantitative RT-PCR of these genes. In both analyses, as noted in response to point G above, *gcs-1* was not induced, while *gst-4* mRNA levels were slightly (2-fold) upregulated upon heat shock (**new Figure S1B**). While results are consistent with a possible engagement of the transcription factor SKN-1 upon heat shock, they did not provide a very strong reason to pursue this idea. In contrast, we observed a strong nuclear translocation of

HLH-30 following heat shock, and we therefore decided to concentrate our efforts on characterizing the role of this factor (Please see response to Reviewer 2, point G below).

Irrespective of our new findings suggesting a role for HLH-30 in the heat-shock response, we now recognize the possible involvement of multiple factors in the discussion on p.15.

“Alternatively, other stress-responsive transcription factors could play roles in inducing autophagy gene transcript levels upon heat shock.”

N. It is known that autophagy inhibition in wt worms limits their lifespan as it is also obvious in some of the curves. The fact that the hormetic effect of heat stress on lifespan is not achieved when autophagy is inhibited could be a result of a general, non-specific detrimental effect of autophagy inhibition on organismal physiology rather than specific inhibition of the hormetic effect.

We and others have found that whole-life autophagy RNAi shortens the lifespan of *C. elegans* irrespective of genetic background, whereas adult-only RNAi of autophagy genes generally does not shorten the lifespan of wild-type animals, as shown in **Table S2** (see also references in Gelino et al, PLOS Genetics, 2016). To better illustrate this, we have added *P*-values to compare autophagy gene RNAi to empty vector controls into the table, similar to the thermo-recovery experiments. Therefore, we do not think that the loss of the hormetic effects in animals subjected to adult-only RNAi of autophagy genes is due to non-specific effects of autophagy inhibition. Instead, our experiments show a specific requirement of autophagy genes for the longevity paradigms of hormesis and HSF-1 overexpression, as has been observed for all longevity paradigms tested so far (reviewed in Lapierre et al., Autophagy, 2015).

Minor points

a. Hormesis refers to the beneficial effects of a mild stressor or treatment that becomes deleterious at higher levels (Gems and Partridge, 2008). The definition given (on page 7) rather refers to antagonistic pleiotropy (Williams, 1957).

We have revised the definition as follows on p. 2, which now says:

“The concept of hormesis refers to a beneficial low-dose stimulation with an environmental agent or exposure to an external stressor that is toxic at a high dose^{13, 14}”

b. Why were HSF-1-overexpressing animals fed bacteria expressing dsRNA against autophagy genes from hatching, while wild-type animals were subjected to RNAi from day 1 of adulthood? (legend of Fig. 2).

We thank the reviewer for pointing this out. We have now replaced this graph with an adult-only RNAi experiment. We have performed the experiments in both ways, i.e., by feeding the animals bacteria expressing dsRNA against autophagy genes from hatching as well as from adulthood. We have found that adult-only autophagy knock-down is sufficient to prevent the increased stress resistance of HSF-1 overexpressing animals. The repeats for all experiments are found in **Table S11**.

c. As the authors report (pages 8 and 9) and Figure 3 shows, hormetic heat shock on day 1 of adulthood caused a significant reduction in the level of intestinal PolyQ44 aggregates on days 2-5. However, according to information provided in the Methods section, it is not clear if the number of aggregates or the mean fluorescence intensity of GFP signal per animal was measured.

We apologize for the confusion on how we measured PolyQ aggregates. We have now dedicated a separate section in the Methods sections on the quantification of the PolyQ aggregates on p. 22, where we say:

“The number of intestinal PolyQ aggregates was counted in individual animals of strains GF80 (Ex(*vha-6p::Q44::yfp* + *rol-6*)) and MAH602 (Is(*vha-6p::Q44::yfp* + *rol-6*)) that were imaged on day 2–5 of adulthood.”

d. Figure S3: I would recommend that the authors explain more explicitly in the text the bafilomycin experiment and how it proves the fact that the heat-shock induced LGG-1 puncta are due to induction of autophagy and not due to inhibition of autosomal turnover. I believe that the samples that need to be directly compared also in the bar graph are number 2 and 3 and number 2 and 4.

We have revised the text to better explain the Bafilomycin A experiment on p.6 (see also response to Reviewer 2, point I below), and have added multiple comparisons of all groups to the graph. For more information on Bafilomycin A experiments in *C. elegans*, please also see Zhang *et al.*, *Autophagy*, 2015 and Wilkinson *et al.*, *Molecular Cell*, 2015, which are both cited in the manuscript.

e. Figure 2B: The title above the graph could change to something more descriptive and not repetitive of the x axis, like "thermotolerance".

We have added section headers for former Figure 2 now **Figure 3**.

f. Authors should use control RNAi in their experiments, to exclude putative nonspecific RNA silencing effects.

The empty vector control that we use in our RNAi experiments is referred to as the L4440 empty vector control, which contains a stretch of about 120 bp, which should engage the RNAi machinery when transcribed. This is used as the standard control in *C. elegans* experiments.

g. Key experiments should be repeated with stable genetic mutants instead of with RNAi lines.

Animals with mutations in autophagy genes are all developmentally compromised. For lifespan analyses and stress assays, it is therefore critical to use adult-only RNAi to exclude interactions with known developmental defects that could impact the experiments. In addition, the available *hsf-1* mutant *hsf-1(sy441)* is not a null mutant and still harbors transcriptional activity (Baird *et al.*, *Science* 2014, Douglas *et al.*, *Cell Reports* 2015).

h. If possible, constitutively active HSF-1 mutants should be examined in addition to HSF-1 overexpressing animals.

We agree that this would be an interesting mutant to analyze, but at this point there is no constitutively active HSF-1 mutant available in *C. elegans*.

i. In Figure 1B-D authors should add representative images of the HSF-1 overexpressing sample as well.

To address this point, and to better address Reviewer 2's point A below on the general organization of the manuscript, we have now moved our autophagy characterization in HSF-1 overexpressing animals to a separate figure (**new Figure 2**). For the resubmission, we have also included images of the GFP::LGG-1-positive punctae in the HSF-1-overexpressing strain, and carried out Bafilomycin A flux assays in these animals (in response also to Reviewer 2's point C below).

j. In figure 3A a positive control (autophagy induction condition) is needed.

We show that PolyQ aggregation is abrogated upon loss of autophagy. Moreover we demonstrate that

HSF-1 overexpression and a hormetic heat shock, which we have shown to induce autophagy, can prevent aggregation in an autophagy-dependent manner. We have now included quantitative RT-PCR results, that demonstrate the induction of autophagy gene mRNA levels upon heat shock in PolyQ-expressing animals (**new Figure S14C-D**), which indicates that autophagy is indeed induced upon heat stress in the PolyQ-expressing animals.

k. The last sentence of the abstract is syntactically incorrect.

We have corrected this sentence in the revised manuscript.

On p. 10, second paragraph, first sentence, I believe it's missing 1 word: ... HSF-1 was sufficient to increase the expression of autophagy-related genes....

This sentence on p.14 reads now as follows:

“We similarly found that HSF-1 overexpression was sufficient to increase the mRNA levels of autophagy-related genes, and also increased the abundance of GFP::LGG-1/Atg8 positive punctae, similarly to heat shock.”

Reviewer #2: Kumsta et al., (Hansen) investigated the interplay between the heat-shock response and autophagy in *C. elegans*. The authors show that a reporter of autophagy and certain autophagy (Atg) genes are induced following heat-shock and in animals overexpressing HSF-1, and that autophagy genes are required for the beneficial effects on stress resistance and lifespan conferred by both HSF-1 overexpression and hormetic heat stress. They also show that hormetic stress reduces the aggregation of polyglutamine in intestinal cells in an autophagy dependent manner.

A. While these observations could be of broad interest to the field, the results are incomplete and will require additional clarification. Perhaps some of this problem is because the paper has two components on the heat shock effects on autophagy and the possible involvement of HSF-1, and somewhat separately the observations on hormesis.

We thank the reviewer for the suggestion to restructure our manuscript and believe that we now better illustrate the novelty and impact of our paper. Specifically, we first describe that a beneficial hormetic heat shock induces autophagy in *C. elegans*, involving, at least in part, transcriptional mechanisms. Secondly, we explore HSF-1 as a possible transcriptional regulator and find that HSF-1 overexpression is sufficient to induce autophagy. We also describe HLH-30, a conserved master regulator of autophagy, as a possible additional factor important for the beneficial effects of heat shock. Third, we demonstrate a direct and functional role for autophagy in hormesis in *C. elegans*, including that a hormetic heat shock early in life protects against PolyQ protein aggregation in an autophagy-dependent manner.

B. The central observation of this paper is that different reporters of autophagy are induced in response to heat shock. While not questioning the statistical significance of induction of the Atg genes upon heat shock, it is notable that the level of induction is quite small relative to the reference heat shock genes that show substantial induction. This brings forth many questions, for example whether this effect is a transcriptional or post-transcriptional response as heat shock also can affect mRNA stability. Of course, the latter observation may be inconsistent with the requirement for HSF-1. Despite the evidence that HSF-1 is involved, it is not possible to conclude from these experiments whether this is a primary consequence of HSF-1 binding to heat shock elements in the promoters of the Atg genes, a secondary consequence of the activity of an HSF-1 directly regulated gene, or that the Atg effect is really HSF-1 independent but heat shock dependent.

We agree with the reviewer's comments on the involvement of HSF-1; for response, please see Reviewer 1, point H above.

C. The effects of heatshock on autophagy shows that GFP::LGG-1/Atg8 forms puncta, however it is unclear how this relates to changes in autophagic activity. Further, how does the induction of a specific subset of Atg genes by heat shock result in increased macroautophagy in a stressed cell? Are these Atg genes that are induced important regulatory components or is it proposed that the entire macroautophagy machinery is upregulated?

We interpret the increase in the number of GFP::LGG-1–positive punctae in response to heat shock as an induction in autophagy, because Bafilomycin A further increased the number of GFP::LGG-1–positive punctae, and because the reduction of autophagy genes with RNAi prevents the formation of GFP::LGG-1–positive punctae. Taken together with the elevated levels of autophagy genes, these observations are consistent with an induction of autophagy in animals subjected to heat shock. It is unclear at this point, however, whether the induction of autophagy upon heat shock is induced by the increased levels of specific autophagy genes that normally could play a limiting role in regulating the autophagy process, or whether other, post-translational mechanisms are involved in increasing autophagy upon heat shock. We note that overexpression of specific autophagy genes has been found to induce autophagy markers and longevity in several different species; i.e., ATG5 overexpression in mice (Pyo *et al.*, Nat Commun. 2013), and Atg8a and Atg8 overexpression in *Drosophila* (Simonsen *et al.*, Autophagy, 2008; Bai *et al.*, PLOS Genetics 2013). It is, however, as of yet unclear how increased expression of specific autophagy genes contributes to overall autophagy induction relevant to specific phenotypes, e.g., lifespan extension in *C. elegans*, as we recently discussed in a review by Lapierre, *et al.*, Autophagy 2015.

D. In looking at Figure 1, the HSF-1 dependence of Atg gene induction could be directly tested either using HSF-1 mutations or RNAi.

Please see response to Reviewer 1, point A above.

E. Could heat-shock or overexpression of HSF-1 affect feeding behavior and thus affect the expression of autophagy genes independently of HSF-1?

We thank the reviewer for raising this important point. We have now performed pharyngeal pumping measurements of the different strains we have used in this study. None of the main strains, including HSF-1– and LGG-1–overexpressing animals show any decrease in pharyngeal pumping (**Figure R5**). We therefore conclude that intrinsic feeding behavior of the animals tested here should not influence the effects of hormesis.

Importantly, it has been reported that heat shock can dramatically reduce pharyngeal pumping (Furuhashi T. *et al.*; Comp Biochem Physiol B Biochem Mol Biol. 2014). We therefore characterized pharyngeal pumping in wild-type animals (N2) upon heat shock and found that a 1 h heat shock dramatically reduced pharyngeal pumping, which was reversible within 30 min upon return to 20°C (**new Figure S10**). To address whether this maximal reduction of 90 min in feeding would have an effect on stress resistance and lifespan, we forced a feeding stop of 90 min by keeping the animals in either liquid M9 media or on bacteria-free agarose plates, followed by thermorecovery and lifespan analyses. We did not find such food-deprived animals to be significantly different from control animals, whereas the heat-shocked animals showed beneficial survival effects (**new Figure S10, Table S1-2**). Taken together, these observations indicate that the short-term food deprivation, caused by diminished pharyngeal pumping upon heat shock are unlikely to lead to the hormetic benefits.

F. While direct ChIP experiments could be done, it would also be useful to search different datasets of HSF-1 chromatin binding in different organisms to see whether there is a direct regulation of Atg

genes and if this is conserved? While answers to these questions would require substantial additional experiments, some of this can also be determined by standard genomics informatics and by generating transgenic lines expressing Atg heat shock inducible gene promoter reporter constructs. Such observations would increase an enthusiasm to embark on the chromatin experiments to demonstrate whether HSF-1 binds directly to Atg gene promoters.

Please see response to Reviewer 1, point H above.

To the specific point about gene promoter reporters, we have now analyzed the expression levels in three different transcriptional reporter strains, i.e. *atg-16.2p::gfp*, *atg-18p::gfp* and *sqst-1p::gfp* and found a significant increase in the fluorescence of these reporter constructs upon heat shock, and show this data in **new Figure S7**.

G. There is also the alternative possibility that HLH-30 has a role in heat-shock induction of Atg genes.

We agree with the reviewer that HLH-30 could be a likely candidate regulator of autophagy during heat shock, since we previously showed HLH-30 to be a conserved regulator of autophagy and important for all longevity paradigms tested in *C. elegans* to date (Lapierre *et al.*, Nat Comm, 2013). We used a translational HLH-30 reporter strain (*hlh-30p::hlh-30::gfp*) to measure the translocation of HLH-30 into the nucleus. Upon heat shock we observed a rapid and reversible translocation of HLH-30 into the nucleus (**new Figure S9A-C**), which indicates that heat shock can activate HLH-30. Consistent with an essential role for *hlh-30* in a beneficial heat-shock response, we also found that *hlh-30* mutants did not benefit from a hormetic heat shock in terms of stress resistance (**new Figure S9E**) and lifespan (**new Figure S9D**). Moreover, analyses of *hlh-30* deletion mutants showed that autophagy gene transcripts *atg-18*, *bec-1/ATG6*, and *lgg-1/ATG8* were not increased upon heat shock and the induction of *sqst-1* was diminished (**new Figure S9D**). Taken together, these results highlight a new role for HLH-30 in regulating autophagy in response to heat shock in *C. elegans*. We have included these data in the manuscript on p. 8-9, where we say:

“In support of the latter possibility, we found that *hlh-30*, the ortholog of mammalian transcription factor EB (TFEB) and a conserved regulator of multiple autophagy-related and lysosomal genes³⁰, was required for the induction of several autophagy genes upon heat shock (**Fig. S9A**). Moreover, hormetic heat shock caused GFP-tagged HLH-30 to rapidly translocate to the nucleus in multiple tissues, including the nerve ring, pharynx, vulva, tail, and intestine (**Figs. S9B-C**), indicating possible activation of HLH-30⁴²⁻⁴⁴. Collectively, these observations suggest a novel role for HLH-30 in regulating autophagy in response to heat shock, similarly to HSF-1. It will be interesting to investigate how the two transcription factors HSF-1 and HLH-30 coordinate autophagy gene expression, and to what extent direct or indirect regulatory mechanisms are involved.”

Of note, we also analyzed the number of GFP::LGG-1/Atg8-positive punctae upon heat shock in various tissues of animals raised on bacteria expressing dsRNA targeting *hlh-30*. Interestingly, we found that both the intestine and hypodermal seam cells (**Figure R6A-B**), but not body-wall muscle (**Figure R6C**), still showed a significant increase in autophagic punctae in response to heat shock. We were only able to count GFP::LGG-1/Atg8-positive punctae upon *hlh-30* reduction using RNAi, because the *hlh-30(tm1978)* mutants expressed GFP::LGG-1/Atg8 at levels that were too low to achieve quality counts. While we were able to show that our *hlh-30* RNAi clone successfully reduced the expression of the HLH-30::GFP marker (**Figure R6D**), we did not quantify *hlh-30* reduction in the experiments in which we counted GFP::LGG-1/Atg8-positive punctae. We have therefore chosen not to include these data in the manuscript. Future experiments are needed to address the possibility that *hsf-1* and *hlh-30* could be differentially required for the process of GFP::LGG-1/Atg8-punctae formation after heat shock; since *hsf-1* appeared to be fully required for autophagosome formation in multiple major tissues upon heat shock whereas *hlh-30* may not be. Such a scenario would be particularly interesting, as this could indicate that an increase in GFP::LGG-1-positive punctae upon

heat shock is not dependent on transcriptional activity of HLH-30.

H. The observations in Figures 1 and S2, the observations on that GFP::LGG-1/Atg8 puncta increase upon heat shock and in Atg mutants? Often puncta of soluble proteins in cells exposed to heat shock is interpreted as protein aggregates. In Figure S2, why does RNAi of Atg genes have little or no effect on foci numbers in the control?

To address the reviewer's important concern whether the GFP::LGG-1-positive punctae represent autophagic events rather than protein aggregation, we performed the following control experiment, using a published strain that expresses mutant LGG-1(G116A) tagged with GFP. The G116A point mutation targets the conserved C-terminal glycine residue that is involved in lipidation and directs LGG-1 to the autophagosomal membrane and renders LGG-1 inactive (Manil-Segalen et al. *Developmental Cell*, 2014). We subjected this reporter strain to our heat shock regimen and analyzed the formation of GFP::LGG-1-positive punctae. We found no or very limited punctae formation in the different tissues, which indicates that heat shock leads to the formation of autophagosomal structures rather than GFP::LGG-1 protein aggregation. We have included these data as **new Figure S2 and new Table S4**, and we now include a comment to this point in the manuscript on p. 5, where we say:

"These punctae represented autophagosomal structures and not heat shock-induced GFP aggregates since we did not observe punctae formation in any tissues upon heat shock in a reporter strain expressing a GFP-tagged LGG-1/Atg8 protein, in which the conserved C-terminal glycine residue (G116) involved in lipidation and targeting of LGG-1 was mutated (**Fig. S2 and Table S4**)³⁸."

We interpret that the reduction of autophagy genes under basal conditions did not significantly change the number of GFP::LGG-1-positive punctae as the result of sufficient residual levels of autophagy genes to sustain autophagy at a basal level; we made a similar argument about the lack of effects on wild-type *C. elegans* lifespan following adult-life autophagy inhibition (see Reviewer 1, point N above and Gelino et al., *PLOS Genetics*, 2016, Zhang et al; *Autophagy*, 2015).

I. While the use of Bafilomycin A treatment, by direct injection into animals, can be interpreted that the observed increase in lgg-1 foci is due to the increased generation of autophagosomes, further experiments are necessary to characterize autophagy flux during heat-shock and the role of HSF-1.

To our knowledge, the Bafilomycin A experiment is currently the most conclusive assay to assess autophagic flux in adult *C. elegans* (Zhang et al; *Autophagy*, 2015). As in other model systems, flux assays using Bafilomycin A test whether a change in autophagic punctae represents a bona fide induction or a block (Klionsky et al., *Autophagy*, 2016). We developed this assay for use in *C. elegans* (Wilkinson et al., *Molecular Cell*, 2015), and we applied it here to evaluate whether the differences in LGG-1 punctae numbers we observed in animals subjected to heat shock were representing an induction, or a block of autophagy. We agree that this same level of rigor should be applied to the HSF-1-overexpressing animals, and we have therefore performed Bafilomycin A injections in these animals as well, and included in the resubmitted manuscript (**new Figure 2E-F**). Our obtained results are consistent with our interpretation that HSF-1-overexpressing animals have increased autophagic activity.

J. Some of the data on tissue-specific analyses of autophagy are confusing. For example, the data presented in Table S1 are not consistent between the tissues (for example only day 1 was analyzed for muscle), certain time points presented in Fig. S4 were analyzed only once according to Table S1 or are even absent from the Table, and in Fig. S4 the scale representing the experiment is not clear. Is the time point 0h before the HS or after? This should be clarified.

We thank the reviewer for pointing this out. We have corrected former Table S1 now **Table S3** to indicate which graph is shown in **Figure S5**. In the table, we show the pre-heat shock measurement in the “Control” column and “0 h” refers to the recovery time, which means this measurement was made immediately following the heat shock. In time-course experiments, in which we used a separate control group that was analyzed at several time points throughout the recovery period, we have now added a row indicating “Pre-HS”. In the graphs for the time-course experiments in **Figure S5**, we have chosen to display time point “0 h” as the beginning of the heat shock. For further clarification, we have added colored shading to the graphs in **Figure S5** to better illustrate the duration of the heat shock.

K. The authors draw conclusions about tissue-specific dynamics of autophagy induction upon heat-shock, however, some of these results could also come from the experimental design. For example, the promoters used to drive the expression of the *lgg-1::GFP* protein fusion could result in differential tissue expression (for example in neurons), rendering the results difficult to compare. Likewise, could the different kinetics observed be due to differential tissue sensitivity to the effects of heat shock?

We agree with the reviewer on this important point. To address this comment and to better discuss the tissue-specific differences that we observe, we have added a paragraph to the discussion section. Please see response to Reviewer 1, point D above.

L. It is shown that Atg genes are required for the beneficial effect of hormetic stress on survival to heat shock at day 4 of adulthood. However, in Table S5, Atg gene RNAi seems to also have an effect on the survival of the control animals at day 4. In these results what are the p-values comparing? As the conclusion of these experiments would seem more central, perhaps they should be in Figure 2.

As noted above in response to Reviewer 1, point E, the adult-only reduction of autophagy genes does not significantly reduce the thermo-recovery of wild-type animals. These data can now be found in **Table S1**. To better illustrate this, we have added *P*-values to compare autophagy gene RNAi to empty vector controls into the table.

M. Did the authors try the hormetic stress at other days of adulthood to assess the protective effect?

Please see response to Reviewer 1, point F above.

N. Perhaps slightly outside the central theme of heat shock and autophagy, have the authors considered the mechanism of hormetic stress mediated stress resistance? In addition to expected roles of HSF-1, ATFS1, XBP1, ATF-6 are other stress factors required? How about HLH-30?

As noted in response to Reviewer 1, point M, we analyzed whether our heat shock regimen could induce other stress response genes, and for this analysis we included target genes of SKN-1, DAF-16, XBP-1/ATF-6, and ATFS-1, and measured their transcript levels using quantitative RT-PCR. Our heat shock only mildly increased the expression of the mitochondrial stress gene *hsp-6*, and oxidative stress gene *gst-4* (**new Figure S1B**). These results would be consistent with a limited engagement of the other transcription factors upon heat shock. In contrast, we observed a strong nuclear translocation of HLH-30 following heat shock and we therefore decided to concentrate our efforts to further characterize the involvement of this factor.

O. Directly relevant to the hormesis observations, are heat shock and autophagy genes more highly induced in animals pre-treated with heat shock versus control animals upon heat-shock at D4? If so, is this due to better HSF-1 or HLH-30 binding to Atg gene promoters?

Please see response to Reviewer 1, point F above.

As noted elsewhere, we agree with the reviewer that it will be interesting to determine in future studies

whether HSF-1 or HLH-30 can bind to promoter regions of distinct sets of autophagy genes and how they coordinate a transcriptional response of autophagy genes upon heat shock.

P. It is an interesting observation that a hormetic heat shock exposure prevents the aggregation of polyQ in intestine and that autophagy genes are required. What is the effect on the biochemical state of polyQ aggregates? If this is linked to autophagy one would expect to see degradation and therefore reduced levels and not a conversion to soluble protein as would occur by enhancing chaperones. Is autophagy directly involved in clearance of polyQ aggregates or is this a chaperone-dependent process of the chaperone disaggregases?

While we agree with the reviewer that it will be very interesting to biochemically characterize how autophagy prevents PolyQ aggregate formation, we argue that this would be better addressed in a detailed follow-up study. We have included a comment to this effect in the result section on p. 16, where we now say:

“The mechanisms by which autophagy limits PolyQ aggregation remain to be elucidated. One possibility could be that increased sequestration of soluble PolyQ proteins limits aggregation formation, in contrast to a possibly conversion of aggregates back to a soluble state. Biochemical analyses of the state of PolyQ aggregates are needed to address this question. Another possibility could be that aggregated PolyQ proteins are turned over by autophagic degradation. Therefore it will be of interest to identify the cargo of autophagic turn-over upon heat stress and in PolyQ-expressing animals. The autophagy-dependent rescue of PolyQ aggregation upon hormetic heat shock is particularly interesting, as this could be of therapeutic potential for the treatment or prevention of diseases with PolyQ expansions.”

Q. Further, given that polyQ expressing animals are impaired, is autophagy itself induced in polyQ expressing animals?

Please see response to point R below.

R. Finally, as polyQ models are available in other tissues, is the hormetic stress and autophagy enhancement protective in these other models?

To this point, we have now tested the effects of a hormetic heat shock on animals expressing neuronal PolyQ40 proteins and found these animals to have reduced aggregation formation. The benefits of the hormetic heat shock on limiting neuronal PolyQ aggregation required autophagy genes, similar to the intestine.

To further analyze the hormetic benefits on PolyQ-expressing animals, we have included lifespan analyses of these animals after hormetic heat shock on day 1 of adulthood. Animals expressing intestinal PolyQ proteins as well as animals expressing neuronal PolyQ proteins have increased longevity after a hormetic heat shock (**new Figure 4E, new Figure S13C and new Table S13**, data described on p. 12 of manuscript).

We have also measured the transcript levels of autophagy genes in animals expressing PolyQ either in the intestine or in neurons on day 1 and day 3 of adulthood. Of note, animals expressing intestinal Q44 proteins already have some aggregates on day 1, whereas animals expressing neuronal Q40 proteins only begin to form aggregates on day 3 and later. Consistently, we observed that intestinal PolyQ aggregation mildly induced the mRNA levels of heat shock protein genes and also of autophagy-related genes, even though these did not reach significance (**new Figure S14A-B**, data on page 12). Neither heat-shock protein genes nor autophagy gene levels were induced in animals expressing neuronal PolyQ proteins, which could be due to the limited aggregation in these animals at these time points (**new Figure S14A-B**, data described on p. 12). Thus, and in response to point Q above, the increased levels of heat-shock protein genes in animals with intestinal PolyQ aggregation

suggests that intestinal PolyQ aggregation induces a HSR, and could also induce autophagy.

Importantly, we observed that a hormetic heat shock on day 1 of adulthood in both PolyQ-expressing models induced autophagy gene transcripts (**new Figure S14C-D**). These findings are consistent with animals living longer because of an increase in autophagy gene levels induced by a hormetic heat shock.

Rebuttal Figures

Figure R1

Figure R1. Reduction of *hsf-1* leads to transcriptional changes of autophagy-related genes. Transcript levels of autophagy genes in wild-type (WT, N2) animals raised from hatching (WL: whole-life RNAi) on control bacteria (empty vector, CTRL) or bacteria expressing dsRNA targeting *hsf-1* maintained under control conditions (CTRL) or WT animals subjected to heat shock for 1 h at 36°C (HS). Data are the mean \pm SEM of four biological replicates, each with three technical replicates, and are normalized to the mean expression levels of four housekeeping genes. Error bars indicate SEM. ns $P > 0.05$, ** $P < 0.01$, *** $P < 0.001$, **** $P < 0.0001$, by one-way ANOVA per transcript.

Figure R2

Figure R2. Reduction of autophagy genes does not inhibit HSF-1 stress body formation. Nuclear HSF-1 stress body formation in hypodermal cells of animals expressing *hsf-1p::hsf-1::gfp* (MAH365) maintained under control conditions (CTRL) or subjected to heat shock for 1 h at 36°C (HS). Animals were anesthetized using NaN_3 , which leads to some stress body formation. Scale bar: 2 μm . This experiment has been repeated twice with similar results.

Figure R3

Figure R3. Reduction of autophagy genes does not inhibit the heat shock response. (A) Transcript levels of autophagy genes on day 1 of adulthood after whole-life RNAi treatment. (B-C) Transcript levels of genes involved in the HSR in wild-type animals raised from hatching (whole-life RNAi) on control bacteria (empty vector, L4440) or bacteria expressing dsRNA targeting the autophagy genes *unc-51/ATG1*, *atg-18*, *bec-1/ATG6*, *lgg-1/ATG8* maintained under control conditions (CTRL) or subjected to heat shock for 1 h at 36°C (HS). Data are the mean \pm SEM of three biological replicates, each with three technical replicates, and are normalized to the mean expression levels of two housekeeping genes. Error bars indicate SEM. ns or no symbol: $P > 0.05$, * $P < 0.05$, ** $P < 0.01$, **** $P < 0.0001$ by multiple t -test. We have included the results of the quantitative RT-PCR of animals treated with *bec-1/ATG6* and *lgg-1/ATG8* RNAi in **new Figure S4C**.

Figure R4 (“REDACTED”)

Figure R5

Figure R5: Pharyngeal pumping is similar in animals used in this study. (A-B) Pharyngeal pumping at day 1 of adulthood at 20°C (N=12-16 animals). Error bars indicate SD. $P > 0.05$, by one-way ANOVA compared to wild-type (WT, N2) animals. This experiment has been repeated three times with similar results.

Figure R6

Figure R6: Heat shock induces GFP::LGG-1/Atg8-positive punctae in *hlh-30*(RNAi) animals. (A-C) GFP::LGG-1/Atg8 punctae were counted in wild-type *C. elegans* (DA2123) raised from hatching (WL: whole-life RNAi) on control bacteria (empty vector, CTRL) or bacteria expressing dsRNA targeting *hlh-30*. Animals were maintained under control conditions (CTRL, solid bars) or subjected to heat shock for 1 h at 36°C (HS, striped bars) followed by the indicated recovery period (Rec). Punctae were examined in (A) hypodermal seam cells (N=95-112 cells), (B) proximal intestinal cells (N=9-15 animals) and (C) body-wall muscle (N=11 animals). Error bars indicate SEM. ns: $P > 0.05$, ** $P < 0.01$, *** $P < 0.001$, **** $P < 0.0001$ by two-way ANOVA. These experiments were repeated three times with similar results. (D) Fluorescence intensity was assessed in animals expressing *hlh-30p::hlh-30::gfp* (MAH235) raised from hatching (WL: whole-life RNAi) on control bacteria (empty vector, CTRL) or bacteria expressing dsRNA targeting *hlh-30* on day 1 of adulthood. Scale bar: 200 μm . Error bars indicate SEM. * $P < 0.05$, **** $P < 0.0001$ by Student's *t*-test (N=10-13 animals). This experiment was performed three times with similar results.

Reviewer #3 (Remarks to the Author)

The authors have made a diligent effort to address the concerns raised during initial review. The overall technical quality of the data are solid, and although the results are somewhat predictable (based on other data regarding heat shock and autophagy induction), the work does represent the first demonstration, to my knowledge, that hormetic heat shock and/or HSF-1 overexpression induces autophagy in *C. elegans* and that such an autophagy response contributes to stress responses, life span, and polyglutamine aggregate clearance. The manuscript falls short in terms of proving the mechanism by which hormetic heat shock induces autophagy (some underdeveloped evidence is provided to support the claim that HSF-1 and/or HLH-30 may be partially responsive) and does not provide any direct evidence that the autophagy-dependent hormetic heat shock beneficial physiological effects (e.g. stress resistance, lifespan extension, protein aggregation) are through HSF-1 or HLH-30.

Given the novelty and interest of the underlying physiology/biology described in this paper, at this point in the review process, my recommendation is that the authors make major further text revisions to focus the paper on what they do show and not make any claims as to what they do not convincingly demonstrate. For example, I think the title should be changed to something like "Hormetic heat stress and HSF-1 improve survival and proteostasis in *C. elegans* via autophagy" without concluding that hormetic heat stress mediates its autophagy-dependent effects via HSF-1. The abstract and text should be changed accordingly. The data on HSF-1 and HLH-30 regulation of autophagy genes could either be deleted entirely or still be included in supplementary materials, but only as evidence to say that the mechanism is not yet known, not to claim hormetic heat shock may be mediated partially and either directly or indirectly through HSF-1 and partially through TFEB. This aspect of the data is simply not well-enough developed to draw any conclusions.

Specific Comments:

1. Page 4. Last sentence of introduction. This paragraph is internally inconsistent. Early in the paragraph, the authors state that heat shock induces autophagy, at least in part via HSF-1, and potentially other transcriptional regulators such as HLH-30. However, in the final sentence they say that "these observations are important because they indicate hormetic heat stress, via HSF-1 regulated autophagy, as a novel mechanism to enhance proteostasis, possible also in age-related protein-folding diseases." The authors data do not definitively support HSF-1 regulated autophagy as the mechanism by which hormetic heat stress induced these effects; i.e. even if their data show an autophagy dependency of hormetic heat stress is it not clear that it is HSF-1-regulation that is critical for the autophagy.

A minor point mentioned during previous reviews is that there is a grammatical problem with this last sentence. Perhaps they mean "is a novel mechanism" rather than "as a novel mechanism". Finally, the use of the term "novel" should be avoided in primary scientific publications.

2. The experiments which show that *hsf-1* RNAi increases autophagy in seam cells (Figure S8A) in control conditions is a problem for the conclusion that heat shock induces autophagy through a mechanism requiring HSF-1. The authors states that this could be due to a block in autophagosomal turnover and that further experiments with bafilomycin A1 would be required to address this. The bafilomycin A1 experiments are straightforward to do and have been done elsewhere by the authors; it is not clear why they were not done in this figure. Also, why would turnover be selectively impaired in seam cells but not in other tissues examined? While there are often tissue-specific effects of different pathways, I think the seam cell data, which the authors acknowledge is an exception to the conclusion that *hsf-1* is required for the heat shock-dependent increase of autophagosomes, is an important enough exception to meaningfully detract from the strength of the overall conclusion that heat shock-induced autophagy is regulated by HSF-1.

3. The authors acknowledge that they do not know whether HSF-1 regulates autophagy directly or indirectly through other transcriptional regulators and therefore, examine the effects of the worm ortholog of TFEB, hlh-30, on heat shock-induction of autophagy gene expression (Fig. S9). However, this aspect of the revision is underdeveloped. While they show some effects of hlh-30 knockdown on autophagy gene expression, they should have also examined whether hlh-30 knockdown blocks heat shock-induced autophagy in the four tissues examined in Figure 1 and 2 (seam cells, muscle, nerve ring, intestine). Is HLH-30 the transcription factor required for heat-shock induced autophagosome formation in the seam cells and other tissues?
4. The authors have not performed any experiments to directly assess whether HSF-1 or HLH-30 are required for the physiological effects of hormetic heat shock on stress resistance, lifespan, or proteostasis. The lack of such data, coupled with the incomplete and/or confusing data on HSF-1 and HLH-30 in autophagy, make it difficult to conclude anything about the role of HSF-1 or HLF-30 in mediating the autophagy-dependent effects of hormetic heat shock in *C. elegans*.
5. The data in Table S13 are difficult for a general readership to decipher. The authors should include the statistical analyses directly in Figure 4 of the lifespan experiment shown in Figure 4 analyzing whether heat shock extends lifespan.

Reviewer #4 (Remarks to the Author)

The authors have been very conscientious in responding to the reviewers' criticisms, and with respect to Reviewer 2 have addressed all of the substantial concerns save the question of whether HSF-1 regulates autophagy genes directly. While the lack of an answer in this case leaves a mechanistic gap, this does not detract from the overall significance and impact of the work. In addition, the paper has been strengthened by addition of other new findings.

I have only two comments: (1) both HSF-1 and HLH-30 are said to have "novel" roles here. How novel can the HLH-30 function be, given that it is already known to be a conserved regulator of autophagy? It is better to let the work speak for itself. (2) in places the writing is a bit flabby and could benefit from some tightening.

Response to reviewers' comments to resubmitted manuscript NCOMMS-16-10042A

We enclose our slightly revised manuscript entitled "Hormetic heat stress and HSF-1 induce autophagy to improve survival and proteostasis in *C. elegans*" by Kumsta et al., and address new Reviewer 3 and 4's comments to our resubmitted manuscript point-by-point below (in blue text).

Reviewer #3: The authors have made a diligent effort to address the concerns raised during initial review. The overall technical quality of the data are solid, and although the results are somewhat predictable (based on other data regarding heat shock and autophagy induction), the work does represent the first demonstration, to my knowledge, that hormetic heat shock and/or HSF-1 overexpression induces autophagy in *C. elegans* and that such an autophagy response contributes to stress responses, life span, and polyglutamine aggregate clearance. The manuscript falls short in terms of proving the mechanism by which hormetic heat shock induces autophagy (some underdeveloped evidence is provided to support the claim that HSF-1 and/or HLH-30 may be partially responsive) and does not provide any direct evidence that the autophagy-dependent hormetic heat shock beneficial physiological effects (e.g. stress resistance, lifespan extension, protein aggregation) are through HSF-1 or HLH-30. Given the novelty and interest of the underlying physiology/biology described in this paper, at this point in the review process, my recommendation is that the authors make major further text revisions to focus the paper on what they do show and not make any claims as to what they do not convincingly demonstrate. For example, I think the title should be changed to something like "Hormetic heat stress and HSF-1 improve survival and proteostasis in *C. elegans* via autophagy" without concluding that hormetic heat stress mediates its autophagy-dependent effects via HSF-1. The abstract and text should be changed accordingly. The data on HSF-1 and HLH-30 regulation of autophagy genes could either be deleted entirely or still be included in supplementary materials, but only as evidence to say that the mechanism is not yet known, not to claim hormetic heat shock may be mediated partially and either directly or indirectly through HSF-1 and partially through TFEB. This aspect of the data is simply not well-enough developed to draw any conclusions.

We agree with the reviewer that we have not fully elucidated the mechanism by which a mild heat shock can induce autophagy in *C. elegans*; to this point, we have changed the title and abstract and have revised the manuscript to better reflect the limitations of our data. That said, we feel that we have accurately discussed the role of *hsf-1* and *hlh-30* as possible mediators of the hormetic effects on survival and proteostasis, and we prefer to keep these data sets in the manuscript (as also suggested by the editor).

Specific Comments:

1. Page 4. Last sentence of introduction. This paragraph is internally inconsistent. Early in the paragraph, the authors state that heat shock induces autophagy, at least in part via HSF-1, and potentially other transcriptional regulators such as HLH-30. However, in the final sentence they say that "these observations are important because they indicate hormetic heat stress, via HSF-1 regulated autophagy, as a novel mechanism to enhance proteostasis, possible also in age-related protein-folding diseases." The authors data do not definitively support HSF-1 regulated autophagy as the mechanism by which hormetic heat stress induced these effects; i.e. even if their data show an autophagy dependency of hormetic heat stress is it not clear that it is HSF-1-regulation that is critical for the autophagy.

We have revised our manuscript and have deleted "via HSF-1 regulated autophagy". The last paragraph of the introduction on page 4 now says:

"Here, we sought to elucidate the molecular mechanisms underlying the beneficial effects of hormetic heat stress by investigating the interplay between heat shock, HSF-1, and autophagy in *C. elegans*. Hormetic heat shock and HSF-1 overexpression induce autophagy in multiple tissues of *C. elegans* and autophagy-related genes are essential for both heat shock-induced and HSF-1-mediated stress resistance and longevity. Finally, we find that hormetic heat shock also improves several models of protein aggregation in an autophagy-dependent manner. These observations are important because they indicate that autophagy

induction by hormetic heat stress is an important mechanism to enhance proteostasis, possibly also in age-related, protein-folding diseases.”

A minor point mentioned during previous reviews is that there is a grammatical problem with this last sentence. Perhaps they mean “is a novel mechanism” rather than “as a novel mechanism”. Finally, the use of the term “novel” should be avoided in primary scientific publications.

The last sentence previously read: “These observations are important because they indicate hormetic heat stress, via HSF-1-regulated autophagy, as a novel mechanism to enhance proteostasis, possibly also in age-related, protein-folding diseases.” As seen above, we have changed this sentence, and we no longer use the word “novel” anywhere in the manuscript.

2. The experiments which show that *hsf-1* RNAi increases autophagy in seam cells (Figure S8A) in control conditions is a problem for the conclusion that heat shock induces autophagy through a mechanism requiring HSF-1. The authors states that this could be due to a block in autophagosomal turnover and that further experiments with bafilomycin A1 would be required to address this. The bafilomycin A1 experiments are straightforward to do and have been done elsewhere by the authors; it is not clear why they were not done in this figure. Also, why would turnover be selectively impaired in seam cells but not in other tissues examined? While there are often tissue-specific effects of different pathways, I think the seam cell data, which the authors acknowledge is an exception to the conclusion that *hsf-1* is required for the heat shock-dependent increase of autophagosomes, is an important enough exception to meaningfully detract from the strength of the overall conclusion that heat shock-induced autophagy is regulated by HSF-1.

We carried out the *hsf-1* RNAi experiment to assay the requirement of *hsf-1* for the increase in autophagosome number after heat shock. We demonstrate that *hsf-1*(RNAi) animals did not display increased levels of autophagosomes after heat shock in any tissue, consistent with HSF-1 regulating autophagy. That said, we observed increased basal levels of autophagy in hypodermal seam cells following reduction of *hsf-1*, which may represent a block, or an induction of autophagy as a compensatory mechanism. We therefore agree with the reviewer that autophagy flux assays will be essential to more accurately evaluate the basal levels of autophagy in hypodermal seam cells upon *hsf-1* depletion in *C. elegans*.

In an unpublished study (Chang *et al.*, manuscript in revision), we show that autophagy in select tissues of *C. elegans* can be differentially regulated, i.e., individual tissues can display a block in autophagy, while other tissues in the same animal may experience an induction of autophagy. Therefore, using the profile of one individual tissue to make arguments about the entire organism may be premature. In our study, we saw equivalent results in three major tissues of the animal, and we now emphasize that our data are consistent with *hsf-1* being required for a heat shock-dependent increase in autophagy at least in these three tissues (page 8):

“RNAi-mediated silencing of *hsf-1* in wild-type animals prevented the heat shock-induced increase in GFP::LGG-1/Atg8 punctae in body-wall muscles, nerve ring neurons, and proximal intestinal cells (hypodermal seam cells were an exception, see Supplementary Fig. 8 and Supplementary Table 5), consistent with *hsf-1* being required for a heat shock-dependent increase of autophagosomes at least in these three major tissues. Collectively, our results suggest that HSF-1 regulates autophagy in *C. elegans*.”

Overall, more experiments are clearly needed to fully understand the complex regulation of autophagy in multicellular organisms, including *C. elegans*.

3. The authors acknowledge that they do not know whether HSF-1 regulates autophagy directly or indirectly through other transcriptional regulators and therefore, examine the effects of the worm ortholog of TFEB, *hlh-30*, on heat shock-induction of autophagy gene expression (Fig. S9). However, this aspect of the revision is underdeveloped. While they show some effects of *hlh-30* knockdown on autophagy gene expression, they should have also examined whether *hlh-30* knockdown blocks heat

shock-induced autophagy in the four tissues examined in Figure 1 and 2 (seam cells, muscle, nerve ring, intestine). Is HLH-30 the transcription factor required for heat-shock induced autophagosome formation in the seam cells and other tissues?

We indeed performed these experiments and included them in our rebuttal to Reviewer #2 Point G, where we wrote:

“Of note, we also analyzed the number of GFP::LGG-1/Atg8-positive punctae upon heat shock in various tissues of animals raised on bacteria expressing dsRNA targeting *hlh-30*. Interestingly, we found that both the intestine and hypodermal seam cells (**Figure R6A-B**), but not body-wall muscle (**Figure R6C**), still showed a significant increase in autophagic punctae in response to heat shock. We were only able to count GFP::LGG-1/Atg8-positive punctae upon *hlh-30* reduction using RNAi, because the *hlh-30(tm1978)* mutants expressed GFP::LGG-1/Atg8 at levels that were too low to achieve quality counts. While we were able to show that our *hlh-30* RNAi clone successfully reduced the expression of the HLH-30::GFP marker (**Figure R6D**), we did not quantify *hlh-30* reduction in the experiments in which we counted GFP::LGG-1/Atg8-positive punctae. We have therefore chosen not to include these data in the manuscript. Future experiments are needed to address the possibility that *hsf-1* and *hlh-30* could be differentially required for the process of GFP::LGG-1/Atg8-punctae formation after heat shock; since *hsf-1* appeared to be fully required for autophagosome formation in multiple major tissues upon heat shock whereas *hlh-30* may not be. Such a scenario would be particularly interesting, as this could indicate that an increase in GFP::LGG-1-positive punctae upon heat shock is not dependent on transcriptional activity of HLH-30.”

4. The authors have not performed any experiments to directly assess whether HSF-1 or HLH-30 are required for the physiological effects of hormetic heat shock on stress resistance, lifespan, or proteostasis. The lack of such data, coupled with the incomplete and/or confusing data on HSF-1 and HLH-30 in autophagy, make it difficult to conclude anything about the role of HSF-1 or HLF-30 in mediating the autophagy-dependent effects of hormetic heat shock in *C. elegans*.

We indeed performed many of these analyses, see e.g., page 9 for discussion of data in which we directly assessed whether HLH-30 is required for the physiological effects of hormetic heat shock on stress resistance and lifespan:

“Similarly, *hlh-30* was required for the beneficial effects of hormetic heat shock on thermal stress resistance (Supplementary Fig. 9d and Supplementary Table 9), and longevity (Supplementary Fig. 9e and Supplementary Table 10).”

We agree with the reviewer that it will be interesting to determine the role of *hlh-30* in regulating proteostasis.

HSF-1 has previously been implicated in hormesis (Kristensen et al., *Journal of Genetics*, 2003; Fonager et al. *Experimental Gerontology*, 2002), and we confirmed the requirement for *hsf-1* in heat shock-induced stress resistance and longevity in our initial experiments. We have included these data in **Rebuttal Figure R7** (see end of this document). In addition, we note that we show that *hsf-1* is required for heat-shock induced rescue of PolyQ formation in Figure 4b-d and Supplementary Figure 12D.

5. The data in Table S13 are difficult for a general readership to decipher. The authors should include the statistical analyses directly in Figure 4 of the lifespan experiment shown in Figure 4 analyzing whether heat shock extends lifespan.

We have now added the mean lifespan (MLS) and *P*-values of the lifespan experiments depicted in the Figures into the figure legends.

Reviewer #4: The authors have been very conscientious in responding to the reviewers' criticisms, and with respect to Reviewer 2 have addressed all of the substantial concerns save the question of

whether HSF-1 regulates autophagy genes directly. While the lack of an answer in this case leaves a mechanistic gap, this does not detract from the overall significance and impact of the work. In addition, the paper has been strengthened by addition of other new findings.

I have only two comments: (1) both HSF-1 and HLH-30 are said to have "novel" roles here. How novel can the HLH-30 function be, given that it is already known to be a conserved regulator of autophagy? It is better to let the work speak for itself. (2) in places the writing is a bit flabby and could benefit from some tightening.

As noted above, we have deleted the word 'novel' from the manuscript. That said, we note that our study indeed contributes several new insights into HLH-30's role in regulating autophagy, namely that HLH-30 rapidly translocates to the nucleus upon heat shock and that *hlh-30* is required for the stress resistance and longevity induced by hormetic heat shock. It will be interesting to investigate if such environmental changes can engage mammalian TFEB in a conserved fashion.

In response to the reviewer's suggestion to tighten the writing of our manuscript, a professional scientific writer has now edited the manuscript.

Rebuttal Figure R7

Rebuttal Figure R7. *hsf-1* is required for hormetically induced thermal stress resistance and longevity.

(a) Survival of wild-type (WT) animals subjected to hormetic heat shock on day 1 of adulthood and then incubated for 8 h at 36°C on day 4 of adulthood. Animals were fed from day 1 of adulthood with control bacteria (empty vector, CTRL) or bacteria expressing dsRNA targeting *hsf-1* (N = 87-101 animals, n = 4 plates). Error bars indicate s.e.m. ns: $P > 0.05$, * $P < 0.05$ by one-way ANOVA. (b) Lifespan analysis of animals subjected to hormetic heat shock with RNAi-mediated *hsf-1* gene reduction from day 1 of adulthood. WT-CTRL animals (12.4 days, N = 106) compared to WT-HS animals (26.3 days, N = 99): $P < 0.0001$; *hsf-1* RNAi-CTRL (16.8 days, N = 121) compared to *hsf-1* RNAi-HS (14.9 days, N = 114): $P < 0.0001$. Each experiment was carried out once.